# WE-MATH 2.0: A VERSATILE MATHBOOK SYSTEM FOR INCENTIVIZING VISUAL MATHEMATICAL REASONING

**Runqi Qiao**[1*†], **Qiuna Tan**[1*†], **Peiqing Yang**[1], **Yanzi Wang**[3], **Xiaowan Wang**[1],
**Enhui Wan**[1], **Guanting Dong**[1], **Shiqiang Lang**[1], **Sitong Zhou**[1], **Yida Xu**[1],
**Yuchen Zeng**[1], **Jie Wang**[1], **Chong Sun**[2], **Chen Li**[2‡], **Honggang Zhang**[1‡]
[1]BUPT    [2]WeChat Vision, Tencent Inc.    [3]Tsinghua University
qrq@bupt.edu.cn   qiunatan@bupt.edu.cn   chaselli@tencent.com
https://we-math2.github.io/

## ABSTRACT

Multimodal Large Language Models (MLLMs) have demonstrated impressive capabilities across various tasks, but still struggle with complex mathematical reasoning. Existing research primarily focuses on dataset construction and method optimization, often overlooking two critical aspects: comprehensive knowledge-driven design and model-centric data space modeling. In this paper, we introduce **WE-MATH 2.0**, a unified system that integrates a structured mathematical knowledge system, model-centric data space modeling, and a reinforcement learning (RL)-based training paradigm to comprehensively enhance the mathematical reasoning abilities of MLLMs. The key contributions of We-Math 2.0 are fourfold: **(1) MathBook Knowledge System:** We construct a five-level hierarchical system encompassing 491 knowledge points and 1,819 fundamental principles. **(2) MathBook-Standard & Pro:** We develop MathBook-Standard, a dataset that ensures broad conceptual coverage and flexibility through dual expansion. Additionally, we define a three-dimensional difficulty space and generate 7 progressive variants per problem to build **MathBook-Pro**, a challenging dataset for robust training. **(3) MathBook-RL:** We propose a two-stage RL framework comprising: (i) Cold-Start Fine-tuning, which aligns the model with knowledge-oriented chain-of-thought reasoning; and (ii) Progressive Alignment RL, leveraging average-reward learning and dynamic data scheduling to achieve progressive alignment across difficulty levels. **(4) MathBookEval:** We introduce a comprehensive benchmark covering all 491 knowledge points with diverse reasoning step distributions. Experimental results show that MathBook-RL performs competitively with existing baselines on four widely-used benchmarks and achieves strong results on MathBookEval, suggesting promising generalization in mathematical reasoning.

## 1 INTRODUCTION

Large Language models (LLMs) have demonstrated remarkable capabilities across a wide range of tasks (Achiam et al., 2023; DeepSeek-AI, 2025; Jaech et al., 2024; Wan et al., 2024; Trinh et al., 2024; Xin et al., 2024). Building on this foundation, Multimodal Large Language Models (MLLMs) have shown impressive performance in visual question answering (VQA) (Bai et al., 2025b; Zhu et al., 2025; Guo et al., 2025; Qiao et al., 2024c; 2025a; Zhang et al., 2024c), optical character recognition (OCR) (Ye et al., 2023b;a; Wei et al., 2024; Tan et al., 2025), and object detection (Liu et al., 2025b; Ren et al., 2024). However, MLLMs still face difficulties with complex reasoning tasks, particularly in visual mathematical problem-solving, where generalization remains a fundamental challenge (Lu et al., 2023; Zhang et al., 2024a; Wang et al., 2024).

Recent efforts to enhance mathematical reasoning in MLLMs have primarily focused on three directions: dataset construction (Lu et al., 2021a; Zhang et al., 2024b; Shi et al., 2024a; Wang et al.,

---

[*]Equal contribution.
[†]Work done as interns at WeChat, Tencent Inc.
[‡]Corresponding author. zhhg@bupt.edu.cn

Table 1: Comparison of We-Math 2.0 with several representative multimodal mathematical datasets.

| Dataset | Usage | Data Annotation | Knowledge-level Annotation | Principle-level Annotation | Difficulty Levels |
|---|---|---|---|---|---|
| Geometry3K (Lu et al., 2021a) | Training Data & Benchmark | Manual | - | - | - |
| MathV360K (Shi et al., 2024a) | Training Data | Collection | - | - | 4 |
| We-Math (Qiao et al., 2024a) | Benchmark | Manual | ✓ | - | - |
| GeoSense (Xu et al., 2025) | Benchmark | Manual | ✓ | ✓ | - |
| **We-Math 2.0 (Ours)** | **Training Data & Benchmark** | **Manual** | ✓ | ✓ | **8** |

2025), preference optimization (Zhuang et al., 2024; Luo et al., 2025), and reinforcement learning (RL) (Huang et al., 2025; Meng et al., 2025). Foundation approaches aggregated datasets from diverse mathematical domains (Shi et al., 2024b). Subsequent efforts introduce structured supervision formats (e.g., Chain-of-Thought (CoT)) combined with preference optimization to guide step-by-step reasoning (Zhuang et al., 2025). More recently, RL-based studies with curriculum-based training have been employed to further improve model performance on complex reasoning tasks (Huang et al., 2025; Wan et al., 2025). Despite this progress, several fundamental challenges remain:

**(1) Lack of a comprehensive knowledge system:** Existing MLLMs show uneven performance across different subfields of math reasoning. (Lu et al., 2023; Wang et al., 2024) Unfortunately, current datasets suffer from limited coverage of knowledge points and domain diversity, underscoring the necessity of establishing a more systematic knowledge system.

**(2) Lack of model-centric difficulty modeling:** Existing multimodal training datasets primarily perform difficulty annotation based on human learning stages (Meng et al., 2025). However, recent studies (Lei et al., 2024; Qiao et al., 2024b; Lu et al., 2023) reveal that MLLMs do not exhibit learning patterns that align well with these human-defined levels. This highlights the need for a more model-centric approach to modeling data difficulty.

**(3) Lack of emphasis on reasoning generalization:** MLLMs are capable of solving complex problems, but perform poorly on corresponding subproblems (Qiao et al., 2024b) as well as on similar, same-type tasks (Zou et al., 2024). This underscores the current training methods' focus on problem memorization rather than fostering reasoning generalization.

To address these limitations, we introduce **WE-MATH 2.0**, a versatile framework that combines a structured mathematical knowledge system, model-centric data space modeling, and a reinforcement learning-based training paradigm to comprehensively improve MLLM's reasoning capabilities (see Table 1. In detail, we begin by establishing the **MathBook Knowledge System**, a five-level hierarchy comprising 491 knowledge points and 1,819 fundamental principles (see Figure 1). This structure is systematically derived from sources such as Wikipedia and open-source textbooks, refined through hierarchical clustering, and further revised by human experts.

Building on this foundation, we introduce **MathBook-Standard**, a dataset featuring comprehensive annotations at the level of 1,819 knowledge principles, along with carefully curated problems to ensure broad, balanced coverage, particularly in underrepresented mathematical domains. To foster deeper conceptual understanding, MathBook-Standard employs dual expansions: *"multi-images per question"* and *"multi-questions per image"*, enabling diverse problem sets that achieve conceptual flexibility. Crucially, we propose a pivotal three-dimensional difficulty modeling framework that redefines mathematical problem construction. By explicitly modeling "step complexity", "visual complexity" and "contextual complexity", each problem is systematically expanded into seven difficulty levels to form **MathBook-Pro**. This design enables structured, progressive learning for MLLMs, laying a strong foundation for improved reasoning across difficulty levels.

To further enhance MLLMs' general mathematical reasoning ability, we propose **MathBook-RL**, a two-stage reinforcement learning framework for progressive and robust training:

**(1) Cold-Start Fine-tuning:** We first adopt a supervised fine-tuning that guides the MLLM to learn knowledge-oriented CoT reasoning, internalizing it to acquire conceptual understanding and structured problem-solving paradigms. **(2) Progressive Alignment RL:** We propose a curriculum-based RL paradigm. Leveraging the *"one-question-multi-image"* and knowledge-point features in MathBook-Standard, we first align the model's analogical reasoning by introducing an average reward mechanism. Building on this foundation, we progressively train the MLLM on MathBook-Pro and further introduce two dynamic scheduling strategies: **i) Knowledge Increment Scheduling:** When errors occur due to complex reasoning steps, the model is adaptively redirected to relevant

incremental-step samples in MathBook-Standard. **ii) Modality Increment Scheduling:** When errors stem from increased modality complexity, the model is guided through single-modality incremental problems. This targeted curriculum enables effective knowledge transfer across difficulty levels.

To comprehensively evaluate MLLMs' reasoning capability, we introduce **MathBookEval**, a benchmark covering all 491 knowledge points with diverse step distributions. Experimental results show that MathBook-RL performs competitively with existing baselines on four widely used benchmarks and substantially improves generalization and robustness. In summary, our contributions are:

- We propose the **MathBook Knowledge System**, a five-level hierarchical framework with 491 knowledge points and 1,819 fundamental principles, enabling systematic and comprehensive mathematical knowledge supervision.
- We develop **MathBook-Standard** and **MathBook-Pro**, two novel datasets that combine comprehensive step-wise annotation, dual expansions for conceptual flexibility, and a principled three-dimensional difficulty modeling framework for structured and progressive learning.
- We introduce **MathBook-RL**, a two-stage RL framework that integrates structured knowledge supervision and dynamic data scheduling, improving the reasoning capabilities of MLLMs.
- We present **MathBookEval**, a benchmark designed to comprehensively evaluate model reasoning across diverse knowledge points and step distributions. Extensive experiments demonstrate that our approach achieves remarkable performance in both generalization and robustness.

## 2 RELATED WORK

**Visual Mathematical Reasoning.** Recently, visual mathematical reasoning has advanced rapidly (Shi et al., 2024a; Zhang et al., 2024b; Zhuang et al., 2024; Han et al., 2024; Luo et al., 2025). Benchmarks such as MathVista (Lu et al., 2023) and MathVision (Wang et al., 2024) assess overall performance, while MathVerse (Zhang et al., 2024a) and Dynamath (Zou et al., 2024) examine reasoning mechanisms and robustness. Methodologically, progress has been made through visual–textual alignment (Shi et al., 2024a; Zhang et al., 2024b; Wang et al., 2025), step-wise reasoning (Zhuang et al., 2024; Luo et al., 2025), and RL-based optimization (Huang et al., 2025; Zhang et al., 2025; Meng et al., 2025; Chen et al., 2025; Liu et al., 2025a; AI et al., 2025; Wan et al., 2025; Qiao et al., 2025b; Zheng et al., 2025; Yang et al., 2025a; Team et al., 2025a; Hong et al., 2025; Team et al., 2025b), which show promising gains on complex tasks. However, robust and generalizable visual reasoning remains an open challenge. Therefore, we propose a systematic, model-centric knowledge system, integrate it with RL-based alignment and a new dataset, aiming to provide fresh insights for the community.

## 3 WE-MATH 2.0

**Overview.** In this section, we introduce **WE-MATH 2.0**, a unified system designed to advance visual mathematical reasoning in MLLMs, developed from three key aspects: (1) We construct a five-level **MathBook Knowledge System** (§3.1), systematically organizing 491 knowledge points and 1,819 fundamental principles for comprehensive mathematical supervision. (2) We propose a **Multi-Dimensional data construction** pipeline (§3.2), incorporating seed problem construction, variant expansion, and principled three-dimensional difficulty modeling. (3) We introduce **MathBookEval** (§3.3), a benchmark aligned with our knowledge system for systematic evaluation.

### 3.1 MATHBOOK KNOWLEDGE SYSTEM

**System Overview.** We construct a five-level hierarchical **MathBook Knowledge System** organized by the "*Definition–Theorem–Application*" paradigm (Fitzpatrick, 2008). The core is a set of knowledge points $\mathcal{K} = \{k_1, k_2, \ldots, k_N\}$, $N = 491$, spanning primary to university mathematics. Each $k_i$ is associated with a set of fundamental principles $\mathcal{P}_i = \{p_{i1}, \ldots, p_{im_i}\}$, $m_i \in [1, 7]$, where $\mathcal{P} = \bigcup_{i=1}^{N} \mathcal{P}_i$ and $|\mathcal{P}| = 1{,}819$. (Figure 3 illustrate examples of principles within the MathBook Knowledge System, while Figure 8 shows how problems are aligned with the system.)

**Hierarchical Construction via Human-AI Collaboration.** We construct $\mathcal{K}$ through a hybrid process. Human experts first design an initial structure $\mathcal{K}^{\text{human}}$ based on authoritative sources,

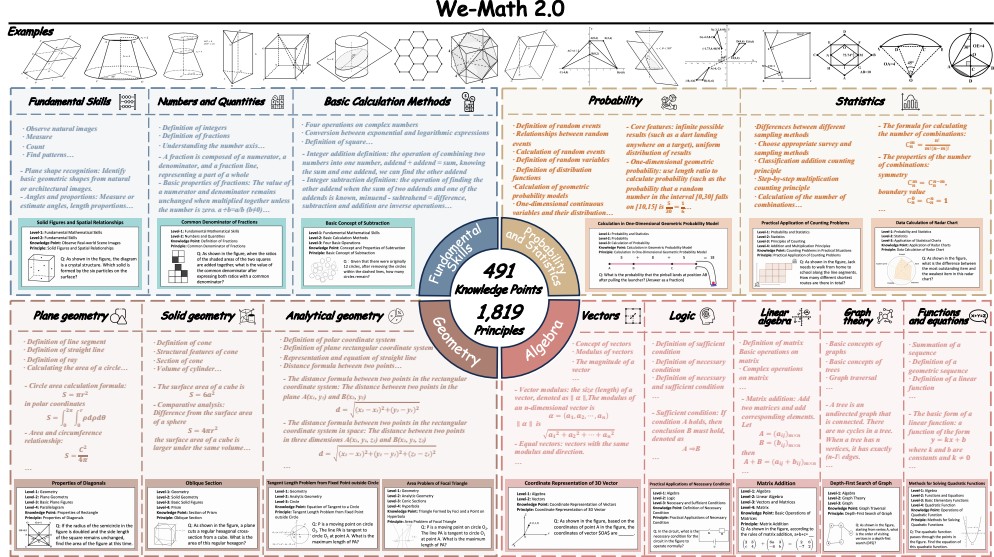

Figure 1: Overview of MathBook, including knowledge points, principles, and sample problems.

including textbooks, Wikipedia, and national curriculum standards. In parallel, we sample 30K problems from the existing math dataset (Lu et al., 2021b; Johnson et al., 2017; Gao et al., 2023; Shi et al., 2024a; Peng et al., 2024; Luo et al., 2025; Zhuang et al., 2024; Zhang et al., 2024b), merging them into a unified dataset. We then use GPT-4o (OpenAI, 2024) to assign multi-level topic tags $\mathcal{T} = \{t_1, \ldots, t_n\}$, followed by hierarchical clustering on the semantic similarity matrix $S \in \mathbb{R}^{n \times n}$ to obtain an AI-generated structure $\mathcal{K}^{\text{auto}}$. The final knowledge point set $\mathcal{K}$ is produced by expert-guided integration of $\mathcal{K}^{\text{human}}$ and $\mathcal{K}^{\text{auto}}$, with independent review for quality assurance.

**Fine-Grained Principle Annotation.** Given the constructed $\mathcal{K}$, we employ GPT-4o to annotate the step-level knowledge points for each problem $q_j \in \mathcal{Q} = \{q_1, \ldots, q_M\}$ by mapping each step in its chain-of-thought solution to the corresponding $k_i \in \mathcal{K}$. This yields a mapping $\mathcal{M}_1 : q_j \mapsto (k_{i_1}, k_{i_2}, \ldots)$, forming a set of step-level solution paths for each knowledge point. Next, for each $k_i$, GPT-4o summarizes the set of theorems and principles used across all associated solution paths, resulting in a mapping $\mathcal{M}_2 : k_i \mapsto \{p_{i1}, \ldots, p_{im_i}\}$. Finally, these AI-extracted principles are consolidated and cross-checked with those written by human experts, with iterative refinement to ensure completeness and accuracy of $\mathcal{P}$. Detailed guideline of our system are listed in Appendix B.1.

## 3.2 MULTI-DIMENSIONAL DATA CONSTRUCTION

In this section, we introduce our data construction pipeline: **MathBook-Standard** & **MathBook-Pro**.

### 3.2.1 MATHBOOK-STANDARD: SEED AND VARIANT PROBLEM CONSTRUCTION

**Seed Problem Construction.** To ensure rich coverage and high-quality design, we construct problems based on the knowledge system following 3 guidelines: **(1)** All diagrams are rendered with GeoGebra for precise geometric representation; **(2)** Problems focus on math essence, avoiding reliance on superficial visual cues; **(3)** Each problem strictly corresponds to its designated principle set $\mathcal{P}_i$. To achieve these, we adopt a "model-assisted, expert-led" workflow. Given a knowledge point $k_i \in \mathcal{K}$ and its associated principle set $\mathcal{P}_i$, an LLM first generates a draft problem, including the question, answer, and XML script. We then use GeoGebra, a software that renders diagrams from XML-based scripts, to automate the generation of draft images: $\mathcal{G}_{\text{LM}}(k_i, p_{ij}) \to (q_i^{\text{draft}}, a_i^{\text{draft}}, x_i^{\text{xml draft}})$. The resulting visual drafts serve as references to guide human experts in constructing problems and diagrams via GeoGebra scripting. In practice, almost all drafts were revised or reworked by experts,[1] in order to avoid reliance on superficial visual cues and ensure proper alignment with the underlying mathematical principles. The final seed problem set is $\mathcal{D}_{\text{seed}} = \{(k_i, p_{ij}, q_i, a_i, I_i, x_i^{\text{xml}})\}$, covering

---

[1]Only 1.2% of the drafts were directly adopted by experts.

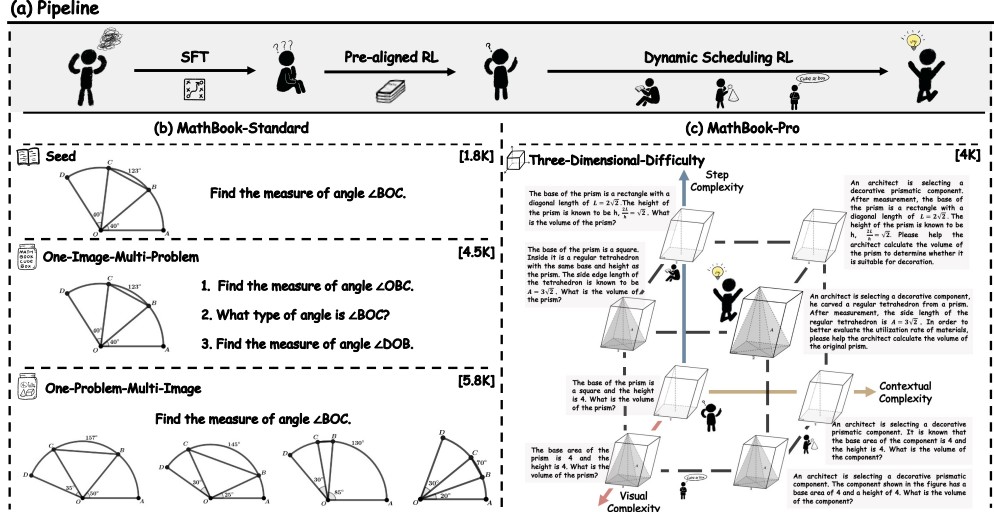

Figure 2: Overview of MathBook dataset and the corresponding training phase.

all knowledge points and principles. The detailed GeoGebra-based diagram generation guidelines can be found in the Appendix B.2.1.

**Variant Problem Expansion.** To further enhance the diversity and generalization ability of the dataset, we systematically construct two types of variants based on each seed problem:

**(1) One-Problem-Multi-Image Variants** ($\mathcal{D}_{\text{ImgVar}}$): Given a seed problem $(q_i, a_i, I_i) \in \mathcal{D}_{\text{seed}}$, we fix the problem statement $q_i$ and knowledge annotation $(k_i, p_{ij})$, and generate a set of images $\{I_i^{(1)}, I_i^{(2)}, \ldots, I_i^{(m)}\}$ by varying the parameters in GeoGebra while maintaining the underlying geometric construction. Each image corresponds to a different geometric instantiation (e.g., acute/obtuse/right triangle), resulting in distinct answers $a_i^{(t)}$: $\{(q_i, a_i^{(t)}, I_i^{(t)}) \in \mathcal{D}_{\text{ImgVar}}, \quad t = 1, \ldots, m\}$. This approach enriches the visual data diversity while preserving semantic consistency.

**(2) One-Image-Multi-Problem Variants** ($\mathcal{D}_{\text{QstVar}}$): Given a seed image $I_i$, we construct multiple new problems $q_i^{(s)}$ targeting different knowledge points $k_i^{(s)}$ and principles $p_{ij}^{(s)}$, curated by experts with language model assistance: $\{((q_i^{(s)}, a_i^{(s)}, I_i) \in \mathcal{D}_{\text{QstVar}}, \quad s = 1, \ldots, n\}$. This strategy leverages the reusability of high-quality diagrams to generate diverse problem variants.

By systematically applying these variant construction methods to each seed problem, we build the **MathBook-Standard** dataset with rich semantic and visual diversity.

### 3.2.2 MATHBOOK-PRO: THREE-DIMENSIONAL DIFFICULTY MODELING

To systematically characterize problem complexity from a model-centric perspective, we define a three-dimensional difficulty space for each seed problem along three orthogonal axes (in Figure 2):

**(1) Step Complexity** ($\phi_s$): Knowledge-oriented reasoning depth is quantified by the number of involved knowledge points, which from the MathBook knowledge system. Given a seed problem with $l$ process-oriented knowledge points, we construct variants requiring $l' > l$ (with at least six).

**(2) Visual Complexity** ($\phi_v$): We increase complexity by adding auxiliary elements (e.g., lines) to the original image via GeoGebra, while preserving the core structure.

**(3) Contextual Complexity** ($\phi_c$): Captures the contextualization of the problem statement. We vary the textual context from concise mathematical descriptions to complex linguistic scenarios.

Each seed problem $(q_0, a_0, I_0) \in \mathcal{D}_{\text{seed}}$ serves as the origin in a structured difficulty space. To enable controlled and interpretable expansion, we generate derived problems by varying a single dimension $d \in \{\phi_s, \phi_v, \phi_c\}$ at a time, yielding variants $(q_i^{(d)}, a_i^{(d)}, I_i^{(d)})$. Through multiple rounds of such single-axis transformations, we progressively construct more complex problems by composing changes

across multiple dimensions. Formally, the most advanced variant takes the form: $(q^*, a^*, I^*) = \phi_s \circ \phi_v \circ \phi_c(q_0, a_0, I_0)$. In **MathBook-Pro**, the expansion along each dimension is implemented as:

**(1)** Along the $\phi_s$ dimension, we introduce intermediate conclusions as new conditions, enabling a knowledge-driven, progressive deepening of reasoning, expressed as $K_{i+1} = K_i + 1$, where $K_i$ denotes the number of knowledge points involved at step $i$. In MathBook-Pro, the most complex step variants involve at least 6 knowledge points.

**(2)** Along the $\phi_v$, we increase visual complexity by adding auxiliary lines, altering geometric configurations or introducing new spatial constructs via GeoGebra, while preserving the core structure.

**(3)** Along $\phi_c$, we embed the mathematical core into real-world contexts or linguistically abstract scenarios, increasing the semantic and contextual demands of the problem statement.

By expanding along the defined dimensions, we generate a set of difficulty-controlled problem variants for each knowledge point, forming the difficulty modeling subset $\mathcal{D}_{\text{difficulty}}$ of MathBook-Pro.

### 3.3 MATHBOOKEVAL

**Design Principles.** To ensure the quality and interpretability of annotations in visual math reasoning tasks, MathBookEval is designed based on the following principles: **(1) Comprehensive Knowledge Coverage:** Problems involve 491 knowledge points, spanning primary to university level, demonstrating broad coverage. **(2) Multi-level Reasoning Depth:** Each problem integrates 1–10 knowledge points, compared to 1–3 (Level 1) in existing process-oriented benchmarks, as illustrated in Figure 27. Notably, our annotation adheres to three principles: (1) integrating public and newly constructed problems under a unified guideline; (2) expert step-by-step annotation with explicit knowledge-point mapping; and (3) independent cross-validation, retaining only consistently annotated items.

**Data Statistics and Evaluation Protocol.** MathBookEval contains **1,000** fully annotated problems, covering all **491** knowledge points in the unified knowledge system $\mathcal{K}$, with 600 problems collected from existing benchmarks and 400 newly curated (Detailed statistics are presented in Table 8). We provide detailed statistics and splits along two key dimensions: **(1) Reasoning Dimension:** Problems are divided by reasoning steps into three levels: 1-3 (Level 1), 4-6 (Level 2), and 7-10 (Level 3), reflecting different reasoning depths. **(2) Knowledge Dimension:** The 491 knowledge points are grouped into 4 domains and 13 subdomains, covering primary to university level. Figure 27 demonstrates superior coverage of knowledge points and reasoning depth. All problems are in multiple-choice or fill-in-the-blank format.

## 4 METHODOLOGY

In this section, we introduce MathBook-RL, a two-stage framework that progressively guides MLLMs to develop reasoning capabilities from easy to hard. The first stage is a cold-start fine-tuning phase that establishes a knowledge-driven reasoning paradigm (§4.1); the second is a dynamic reinforcement learning phase that enhances the model's generalization ability (§4.2).

### 4.1 COLD-START FINE-TUNING

The cold-start supervised fine-tuning (SFT) stage aims to instill explicit awareness of knowledge system and a knowledge-driven reasoning paradigm, avoiding rote memorization. The initial training set $\mathcal{D}_{\text{init}}$ is built from MathBook-Standard, which fully covers all 491 knowledge points. To improve rationale interpretability, we use GPT-4o (OpenAI, 2024) to rewrite each sample with natural language explanations that explicitly reference the relevant knowledge. The model is then trained using standard supervised fine-tuning: $\mathcal{L}_{\text{SFT}}(\theta) = \mathbb{E}_{(x,y)\sim\mathcal{D}_{\text{init}}}\left[-\log P_\theta(y \mid x)\right]$. This stage enhances the model's ability to internalize the knowledge system and follow knowledge-guided reasoning chains.

### 4.2 PROGRESSIVE ALIGNMENT REINFORCEMENT LEARNING

**(1) Pre-aligned RL.** Prior to the dynamic scheduling stage, we perform initial RL training on MathBook-Standard dataset to ensure that the model develops genuine understanding of mathematical knowledge. Specifically, we utilize the $\mathcal{D}_{\text{ImgVar}}$ subset, where each group contains multiple variants

of the same knowledge principle: $(q_i, a_i^{(t)}, I_i^{(t)}) \in \mathcal{D}_{\text{ImgVar}}, \ t = 1, \ldots, m$. To encourage consistent and robust performance across different formulations, we adopt a mean-based reward function: $r = \frac{1}{m} \sum_{t=1}^{m} r^{(t)}$, where $r^{(t)} = 0.9$ if the answer is correct, $0.1$ if only the format is correct, and $0$ otherwise. Specifically, for problems corresponding to the same knowledge principle, rollout rewards are first sorted within each problem. Next, the mean reward at each sorted position is calculated across these problems and subsequently employed in the calculation of $A_i$. Instead of focusing on individual problems, this design integrates rewards across all problems corresponding to the same knowledge principle, thereby providing a more comprehensive critic.

**(2) Dynamic Scheduling RL.** In this section, we introduce a dynamic RL algorithm based on MathBook-Pro. The training process is organized as a curriculum along a main trajectory of increasing difficulty, primarily centered on the knowledge dimension. For each base problem $(q_0, a_0, I_0)$, denoted as $x_0$, we construct a sequence of increasingly challenging variants as follows:

$$x_0 \rightarrow \phi_s(x_0) \rightarrow \phi_s \circ \phi_v(x_0) \rightarrow \phi_s \circ \phi_c(x_0) \rightarrow \phi_s \circ \phi_v \circ \phi_c(x_0) \tag{1}$$

where $\phi_s$ denotes increasing the number of knowledge points, $\phi_v$ and $\phi_c$ denotes increasing visual complexity and contextual abstraction. This forms a progressive path from basic to advanced reasoning for each knowledge anchor.

**Incremental Learning Mechanism.** At each curriculum transition $x \rightarrow \phi(x)$, if the model fails on $\phi(x)$ after succeeding on $x$, we introduce an incremental learning step. Specifically, we define the incremental set $\Delta(x, \phi)$ as a collection of samples that isolate the new knowledge or modality introduced by $\phi$. The model is first trained on $\Delta(x, \phi)$ to address the incremental challenge, then reattempts $\phi(x)$. Concretely:

- **Knowledge Increment Scheduling**: For $x_0 \rightarrow \phi_s(x_0)$, if the model fails on $\phi_s(x_0)$, we construct $\Delta(x_0, \phi_s)$, comprising auxiliary problems $x_0'$ that target the new knowledge point(s) from $\phi_s$.
- **Modality Increment Scheduling**: For $\phi_s(x_0) \rightarrow \phi_s \circ \phi_v(x_0)$ (or $\phi_s \circ \phi_c(x_0)$), if the model fails on the more complex sample, we construct $\Delta(\phi_s(x_0), \phi_v)$ (or $\Delta(\phi_s(x_0), \phi_c)$), which contains samples isolating the new visual or contextual complexity.

This incremental adaptation, denoted by $\Delta(x, \phi)$ at each step, ensures that the model can efficiently bridge the gap between curriculum stages. Notably, our overall RL objective is optimized using Group Relative Policy Optimization (GRPO) (Shao et al., 2024), which extends PPO by estimating the baseline from group scores instead of a separate critic. The GRPO objective is:

$$\mathcal{J}(\theta) = \mathbb{E}[q \sim P(Q), \{o_i\}_{i=1}^{G} \sim \pi_{\theta_{old}}(O|q)] \frac{1}{G} \sum_{i=1}^{G} \frac{1}{|o_i|} \sum_{t=1}^{|o_i|}$$

$$\left\{ \min \left[ \frac{\pi_\theta(o_{i,t}|q, o_{i,<t})}{\pi_{\theta_{old}}(o_{i,t}|q, o_{i,<t})} \hat{A}_{i,t}, \text{clip} \left( \frac{\pi_\theta(o_{i,t}|q, o_{i,<t})}{\pi_{\theta_{old}}(o_{i,t}|q, o_{i,<t})}, 1 - \epsilon, 1 + \epsilon \right) \hat{A}_{i,t} \right] - \beta \mathbb{D}_{KL} \left[ \pi_\theta || \pi_{ref} \right] \right\}, \tag{2}$$

where $\epsilon$ and $\beta$ are hyperparameters, $q$ denotes the input, $\{o_i\}_{i=1}^{G}$ are sampled outputs, and $r_i$ is the corresponding reward. $\hat{A}_{i,t}$ is the normalized advantage value for the $i$-th trajectory in the group. This curriculum-driven RL process, augmented with explicit incremental adaptation at each stage, enables the MLLM to progressively master complex, multi-dimensional reasoning tasks while preserving stability and generalization across knowledge, visual, and contextual variations.

## 5 EXPERIMENTS

### 5.1 EXPERIMENTAL SETUP

**Datasets.** All training data are sourced from WE-MATH 2.0 in compliance with copyright and licensing requirements, and all expert-constructed problems will be released under appropriate CC licenses. We use 1K, 5.8K and 4K samples for SFT, pre-aligned RL, and dynamic scheduling RL stages, respectively. Experiments are conducted on four standard mathematical reasoning benchmarks: MathVista (Lu et al., 2023), MathVision (Wang et al., 2024), MathVerse (Zhang et al., 2024a), and We-Math (Qiao et al., 2024a). Detailed evaluation protocols are provided in Appendix C.5.1.

**Baselines.** We conduct our training based on both Qwen2.5-VL-7B and Qwen2.5-VL-3B, and compare our method with three categories of baselines: **(1) Closed-source models** (e.g., GPT-4o (OpenAI, 2024)); **(2) Open-source general models** (e.g., InternVL2.5 series (Chen et al., 2024a),

Table 2: Performance comparison across four widely-used mathematical reasoning benchmarks. Each benchmark follows its standard evaluation metric: MathVista and MathVision use accuracy, We-Math reports the strict score, and MathVerse evaluates on the vision-only subset with accuracy. Data sizes used for SFT and RL are annotated in blue and red, respectively.

| Model | #Data | Avg. | MathVista | MathVision | We-Math | MathVerse |
|---|---|---|---|---|---|---|
| *Closed-source* | | | | | | |
| GPT-4o-latest | - | **54.0** | **71.6** | **43.8** | **50.6** | **49.9** |
| Gemini-1.5-Pro | - | 53.6 | 67.9 | 41 | 50.5 | 54.8 |
| *Open-source (General)* | | | | | | |
| Qwen2.5-VL-7B | - | **42.6** | **68.2** | 25.1 | 36.0 | **41.1** |
| InternVL2.5-8B-BoN-8 | - | 41.7 | **68.2** | **25.6** | **38.6** | 34.5 |
| *Open-source (Reasoning)* | | | | | | |
| Math-PUMA-7B | *1.88M* | - | 47.9 | - | 19.2 | 26.0 |
| URSA-8B | *2.96M* | 37.8 | 58.8 | 28.7 | 32.8 | 31.0 |
| R1-OneVision-7B | *155K*+*10K* | - | 64.1 | **29.9** | 30.1 | - |
| R1-VL-7B | *260K*+*10K* | - | 63.5 | 24.7 | 22.7 | - |
| MM-Eureka-7B | *15K* | 45.2 | **73.0** | 26.9 | 34.5 | 46.2 |
| WeThink-7B | *120K*+*20K* | 47.5 | 71.6 | 26.0 | **48.0** | 44.2 |
| VLAA-Thinker-7B | *25K* | 46.0 | 68.0 | 26.4 | 41.5 | **48.2** |
| OpenVLThinker-7B | *35K*+*15K* | - | 72.3 | 25.9 | - | - |
| **MathBook-7B (Ours)** | *1K*+*9.8K* | **48.7** | **73.0** | **28.0** | 48.4 | 45.2 |
| Δ (*vs* Qwen2.5-VL-7B) | - | *+6.1* | *+4.8* | *+2.9* | *+12.4* | *+4.1* |

Qwen2.5-VL series (Bai et al., 2025b)); **(3) Open-source reasoning models** (e.g., R1-VL (Zhang et al., 2025)). Our evaluation is based on VLMEvalKit (Duan et al., 2024). Detailed descriptions of baselines are provided in Appendix C.5.2.

## 5.2 MAIN RESULTS

Table 2 displays the performance of our MathBook-7B across various benchmarks. Overall, our method achieves remarkable performance, clearly demonstrating its superiority. Further analysis reveals the following observations. (Results for the 3B setting are provided in Appendix C.2.)

**(1) Overall superiority of MathBook.** Compared to the backbone Qwen2.5-VL-7B, MathBook-7B achieves over a 6% improvement across all benchmarks, validating the effectiveness of our approach.

**(2) Effectiveness of progressive alignment reinforcement learning on knowledge generalization.** Focusing on the We-Math benchmark, which requires solving both complex multi-step questions and their corresponding subproblems, MathBook-7B outperforms strong RL baselines. This demonstrates the effectiveness of progressive alignment reinforcement learning in knowledge generalization.

**(3) Less is More: Efficiency with limited training data.** MathBook-7B achieves strong performance using only 9.8K training samples. We attribute this to the high-quality, structured mathematical knowledge system we constructed, enabling efficient alignment and generalization with limited data.

## 5.3 RESULTS ON MATHBOOKEVAL

To investigate MLLM abilities in reasoning depth and knowledge coverage breadth, we conduct Math-BookEval and observe the following (see Table 3). Complete results are provided in Appendix B.4.6.

**(1) MLLMs performance negatively correlates with the number of required knowledge points.** As reasoning steps increase, model accuracy declines. In particular, problems requiring 7–10 knowledge points yield the lowest accuracy (below 50%). These results highlight the challenge of multi-step reasoning and validate knowledge points as a key measure for modeling problem difficulty.

Table 3: The performance of different MLLMs on MathBookEval for reasoning evaluation. **Acc.**: Accuracy; **FS.**: Foundational skills; **PS.**: Probability and statistics; **Geo.**: Geometry; **Alg.**: Algbra

| Models | Acc. | Reasoning | | | Knowledge | | | |
|---|---|---|---|---|---|---|---|---|
| | | Level1 | Level2 | Level3 | FS. | PS. | Geo. | Alg. |
| *Closed-source MLLMs* | | | | | | | | |
| GPT-4o | 50.8 | 52.8 | 48.9 | 41.7 | 33.8 | 57.6 | 44.2 | 67.2 |
| GPT-4V | 42.8 | 44.0 | 43.0 | 31.9 | 36.8 | 56.6 | 33.5 | 59.4 |
| *Open-source MLLMs* | | | | | | | | |
| InternVL2.5-78B | 51.8 | 52.5 | 51.8 | 45.8 | 50.0 | 64.2 | 42.6 | 67.6 |
| Qwen2.5-VL-72B | 57.1 | 58.3 | 56.4 | 50.0 | 52.9 | 58.5 | 52.1 | 68.8 |
| LLaVA-OneVision-72B | 43.0 | 44.8 | 42.0 | 31.9 | 38.2 | 52.8 | 37.0 | 53.5 |
| InternVL2.5-8B | 37.9 | 40.7 | 34.5 | 27.8 | 33.8 | 46.2 | 31.4 | 50.0 |
| Qwen2.5-VL-7B | 46.7 | 50.1 | 43.0 | 33.3 | 44.1 | 58.5 | 38.8 | 60.2 |
| LLaVA-OneVision-7B | 31.6 | 34.3 | 28.0 | 23.6 | 36.8 | 41.5 | 24.9 | 41.0 |
| R1-VL-7B | 38.0 | 41.9 | 32.6 | 27.8 | 38.2 | 38.7 | 32.3 | 50.4 |
| MM-Eureka-7B | 50.0 | 50.6 | 51.5 | 38.9 | 47.1 | 52.8 | 44.0 | 62.9 |
| **MathBook-7B** | 50.4 | 52.0 | 48.2 | 45.8 | 57.4 | 67.9 | 40.5 | 63.3 |

**(2) MLLMs perform well on algebra but struggle with geometry.** Along the knowledge dimension, most MLLMs demonstrate strong performance in algebra, achieving accuracies above 50%. However, their consistently poor performance in geometry highlights ongoing challenges in spatial reasoning.

**(3) Larger models yield more consistent improvements across all dimensions.** Within the InternVL2.5 and Qwen2.5-VL families, increasing model size leads to consistent gains across all dimensions and in overall scores, emphasizing the role of scale in enhancing reasoning capabilities.

## 5.4 QUANTITATIVE ANALYSIS

**Ablation Study.** As shown in Table 4, we conduct ablation studies on the training stages. M0 denotes MathBook-7B, while M1–M4 represent models at different training stages (RL-Pre: Pre-aligned RL; RL-Dyn: Dynamic Scheduling RL). We lead to following two key findings:

**(i) Both RL stages contribute significantly.** Each RL stage (M0-M3) yields progressive improvements over M4. In particular, pre-aligned reinforcement learning (RL) in the first stage yields impressive results on MathVista and We-Math benchmarks, highlighting the crucial role of knowledge learning in enhancing mathematical reasoning abilities. **(ii) SFT alone offers limited gains, but is crucial for unlocking RL potential.** Comparing M0, M3, M4, we find that SFT alone yields marginal improvements over the Qwen2.5-VL backbone. How-

Table 4: Results of the ablation study. **MVt**: MathVista; **MVs**: MathVision; **WM**: We-Math)

| Method | SFT | RL-Pre | RL-Dyn | MVt | MVs | WM |
|---|---|---|---|---|---|---|
| $M_0$ | ✓ | ✓ | ✓ | 73.0 | 28.0 | 48.4 |
| $M_1$ | ✓ | ✓ | - | 72.4 | 27.0 | 47.2 |
| $M_2$ | ✓ | - | ✓ | 72.0 | 26.3 | 43.3 |
| $M_3$ | - | ✓ | ✓ | 71.5 | 26.3 | 46.7 |
| $M_4$ | ✓ | - | - | 65.8 | 25.7 | 38.3 |

ever, when combined with RL, SFT version substantially boosts overall performance, highlighting its critical role in shifting the model's reasoning paradigm and enabling more effective RL optimization.

**Analysis of SFT Data Paradigm and Scale.** We explore the impact of data paradigm and scale during the SFT stage. Based on the M0 setting, we consider two variants: (1) Replacing the natural language CoT format with a structured, step-wise CoT format (Zhuang et al., 2024) aligned with $\mathcal{K}$; (2) Increasing the SFT data scale with larger datasets (from 1K to 15K).

**(i) Natural language CoT outperforms the structured step-wise format in SFT.** As shown in Table 5, natural language CoT outperforms the structured format in RL. This highlights the advantage of natural language prompts in cultivating flexible reasoning, which in turn strengthens visual mathematical reasoning skills. **(ii) Minimal SFT suffices to unlock RL potential.** Scaling up SFT data does not

Table 5: SFT Data Analysis. **SFT (Str.)** and **SFT (Lar.)** denotes structured and large-scale SFT training.

| Setting | MVt | MVs | WM |
|---|---|---|---|
| $M_0$ | 73.0 | 28.0 | 48.4 |
| SFT(Str.) | 71.9 | 26.0 | 46.7 |
| SFT(Lar.) | 72.8 | 27.0 | 49.0 |

improve performance. Models trained on minimal, well-curated data perform comparably or even better than those trained on larger datasets, suggesting that a small, high-quality SFT set suffices to establish the reasoning paradigm for effective RL.

## 6    CONCLUSION

In this work, we present **WE-MATH 2.0**, a unified framework for multi-modal mathematical reasoning. It comprises: **(1) MathBook Knowledge System**, a five-level hierarchy covering 491 knowledge points and 1,819 fundamental principles for comprehensive supervision; **(2) MathBook-Standard** and **MathBook-Pro**, two richly annotated datasets with conceptual expansions and principled difficulty modeling for structured learning; **(3) MathBook-RL**, a two-stage reinforcement learning framework that leverages knowledge-guided supervision and dynamic data scheduling; and **(4) Math-BookEval**, a benchmark for evaluating reasoning across diverse knowledge and step distributions. Extensive experiments validate MathBook's effectiveness in enhancing generalization of MLLMs.

## REPRODUCIBILITY STATEMENT

**Data.** We ensure that all datasets developed in this work, including *MathBook Knowledge System*, *MathBook-Standard*, *MathBook-Pro*, and *MathBookEval*, will be fully released upon publication. To support reproducibility during the review phase, we provide representative data samples (up to the maximum size allowed by the submission system, 100MB) in the supplementary material.

**Experimental Setup.** The experimental protocol, including dataset sizes, training stages, and evaluation benchmarks, is described in Section 5 and Appendix C.1. In addition, our code contains complete environment specifications (e.g., `requirements.txt`), along with scripts for dataset preparation, training, and evaluation. These files ensure that the reported results can be reproduced on standard hardware with minimal configuration.

**Code and Model Checkpoints.** We include the full code in the supplementary material, together with scripts for data preparation, training, and evaluation. Due to file size limitations, pretrained model checkpoints cannot be submitted at this stage. Upon acceptance, we will release all checkpoints in the camera-ready version to facilitate rapid validation of our results.

**Methodology.** We detail the proposed framework in Section 3 and Section 4. Specifically, we (i) specify the *MathBook Knowledge System* (five-level hierarchy; **491** knowledge points; **1,819** principles), (ii) describe the *MathBook-Standard* pipeline with two orthogonal expansions (one-question–multi-image; one-image–multi-question), (iii) formalize the three-axis difficulty space for *MathBook-Pro* (step, visual, contextual) yielding seven progressive variants per seed, and (iv) present *MathBook-RL*, a two-stage training paradigm.

## ETHICS STATEMENT

**Licensing and Open Access.**    For all referenced or incorporated data in WE-MATH 2.0, we only use existing datasets with clear and appropriate licenses. All data curated by our team will be released under the CC BY 4.0 license, ensuring open access for the research community. The entire MathBook dataset, including both external and newly constructed components, will be made publicly available to facilitate further research and development.

**Data Sources and Privacy.**    All data in WE-MATH 2.0 are either sourced from publicly available datasets or generated by our expert team, and do not contain any personal user information. Therefore, there are no privacy concerns related to our dataset.

**Expert Compensation.**    Experts involved in annotation are compensated on a per-task basis, with payment issued only after cross-validation and quality assurance. All compensation meets or exceeds the local minimum wage standards.

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

# Appendix

CONTENTS

## A  BROADEN IMPACT

**Towards principled and generalizable mathematical model training.**  WE-MATH 2.0 provides a comprehensive and structured mathematical knowledge system, which fills the gap left by previous works that lack a complete and systematic framework. By introducing a five-level hierarchy with 491 knowledge points and 1,819 fundamental principles, MathBook enables more principled and interpretable mathematical learning for MLLMs. The dual expansion strategy ("multi-images per question" and "multi-questions per image") and the three-dimensional difficulty modeling not only enrich the diversity of training data but also facilitate robust and progressive learning. This systematic approach can inspire future research to adopt knowledge-driven and model-centric data space modeling, leading to more reliable and generalizable mathematical reasoning models. Furthermore, the fine-grained annotations and progressive difficulty levels provide a valuable resource for benchmarking and analyzing the strengths and weaknesses of different MLLMs, promoting transparency and interpretability in model development.

**Bridging AI and Education: high-quality datasets for teaching and learning.**  WE-MATH 2.0's datasets are not only designed for model training but also have significant educational value. Each problem is accompanied by a GeoGebra (GGB) file, which can serve as high-quality teaching material for educators and students. The hierarchical knowledge system and step-wise annotations make it easier to design personalized learning paths and targeted exercises, supporting adaptive learning and formative assessment. The multi-modal and multi-perspective problem sets encourage students to develop flexible thinking and deepen their conceptual understanding. By bridging the gap between AI research and educational practice, MathBook has the potential to enhance mathematics education, facilitate interactive and engaging learning experiences, and support the development of intelligent tutoring systems.

**Enhancing RL generalization through progressive and dynamic training.**  WE-MATH 2.0 introduces a novel, model-centric curriculum for RL-based training, where problems are systematically organized from easy to hard based on explicit difficulty modeling. This approach provides a new perspective for designing RL curricula, enabling more effective and efficient learning. The "one-question-multi-image" and "one-image-multi-question" strategies, together with dynamic scheduling mechanisms, enhance the robustness and generalization of RL-trained models. These innovations can inspire the broader RL community to explore curriculum learning, dynamic data scheduling, and multi-modal data augmentation for complex reasoning tasks. Moreover, these hierarchical knowledge approaches also offers a new solution for tool learning. MathBook thus serves as a valuable testbed for advancing RL methods in the context of mathematical reasoning and beyond.

## B  DETAILS OF WE-MATH 2.0

### B.1  MATHBOOK KNOWLEDGE SYSTEM

#### B.1.1  HIERARCHICAL STRUCTURE OF KNOWLEDGE POINTS

As illustrated in Figure 3, we provide an overall view of the hierarchical structure of knowledge points in the **MathBook Knowledge System**. Figures 4–7 further present partial examples of different substructures at varying depths of the hierarchy. The system is organized as a five-level hierarchical structure of knowledge points $\mathcal{K} = \{k_1, k_2, \ldots, k_N\}$, where $N = 491$ denotes the total number of knowledge points at the lowest level of the hierarchy. The first level consists of four categories: *Geometry*, *Fundamental Skills*, *Algebra*, and *Probability and Statistics*.

The construction of $\mathcal{K}$ follows a two-track, human-AI collaborative process. First, an initial version $\mathcal{K}^{\text{human}}$ is constructed by collecting and merging knowledge point lists from authoritative sources, including Wikipedia, open-source mathematics textbooks, and national curriculum standards. This initial structure is deduplicated, reorganized, and refined for logical consistency and comprehensive coverage.

In parallel, a large-scale problem set $\mathcal{Q}$ is collected, including 30,000 sampled from existing math datasets. GPT-4o is used to assign multi-level topic tags $\mathcal{T} = \{t_1, \ldots, t_n\}$ to each problem, and a

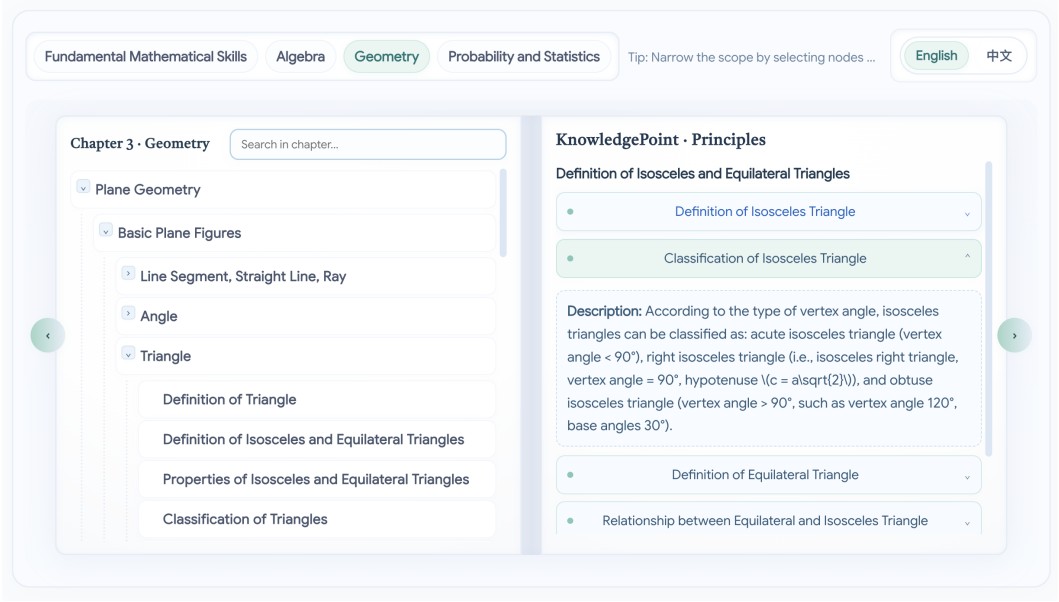

Figure 3: Overall view of the hierarchical structure of knowledge points in MathBook.

semantic similarity matrix $S \in \mathbb{R}^{n \times n}$ is computed. Hierarchical clustering is then applied to $S$ to generate an AI-assisted hierarchical structure of knowledge points $\mathcal{K}^{\text{auto}}$.

Finally, the AI-assisted structure $\mathcal{K}^{\text{auto}}$ is integrated with the initial structure $\mathcal{K}^{\text{human}}$ through systematic comparison, merging, and revision. The $\mathcal{K}^{\text{auto}}$ serves as a reference for revising and refining the manually constructed hierarchical structure of knowledge points, resulting in the final knowledge point set $\mathcal{K}$.

### B.1.2  KNOWLEDGE PRINCIPLES

As shown in Figures 9–12, we provide several examples of knowledge principles, which include definitions, theorems, and other foundational statements associated with each knowledge point. The annotation of principles $\mathcal{P} = \bigcup_{i=1}^{N} \mathcal{P}_i$ ($|\mathcal{P}| = 1{,}819$) also follows a two-track, human-AI collaborative approach.

Based on the constructed knowledge hierarchy $\mathcal{K}$, a set of core principles for each knowledge point $k_i$ is first drafted, referencing authoritative sources such as Wikipedia, textbooks, and international curriculum standards.

In parallel, for each $k_i$, a set of representative problems from $\mathcal{Q}$ is selected and their chain-of-thought (CoT) solutions are annotated. Each step in the CoT is mapped to the corresponding knowledge point using GPT-4o, and the relevant CoT steps for each $k_i$ are extracted. The CoT steps associated with each knowledge point are then aggregated and reviewed to supplement, refine, and validate the set of principles for $k_i$.

This process is repeated iteratively, consolidating both the expert-written and data-driven principles, cross-checking against original sources and annotated solution paths, until the set of principles $\mathcal{P}_i$ for each knowledge point is comprehensive and precise.

This process is repeated iteratively, consolidating both the expert-written and data-driven principles, cross-checking against original sources and annotated solution paths, until the set of principles $\mathcal{P}_i$ for each knowledge point is comprehensive and precise. As illustrated in Figure 8, we further provide an example from **We-Math 2.0**, where each problem is explicitly aligned with its corresponding knowledge point and associated principle.

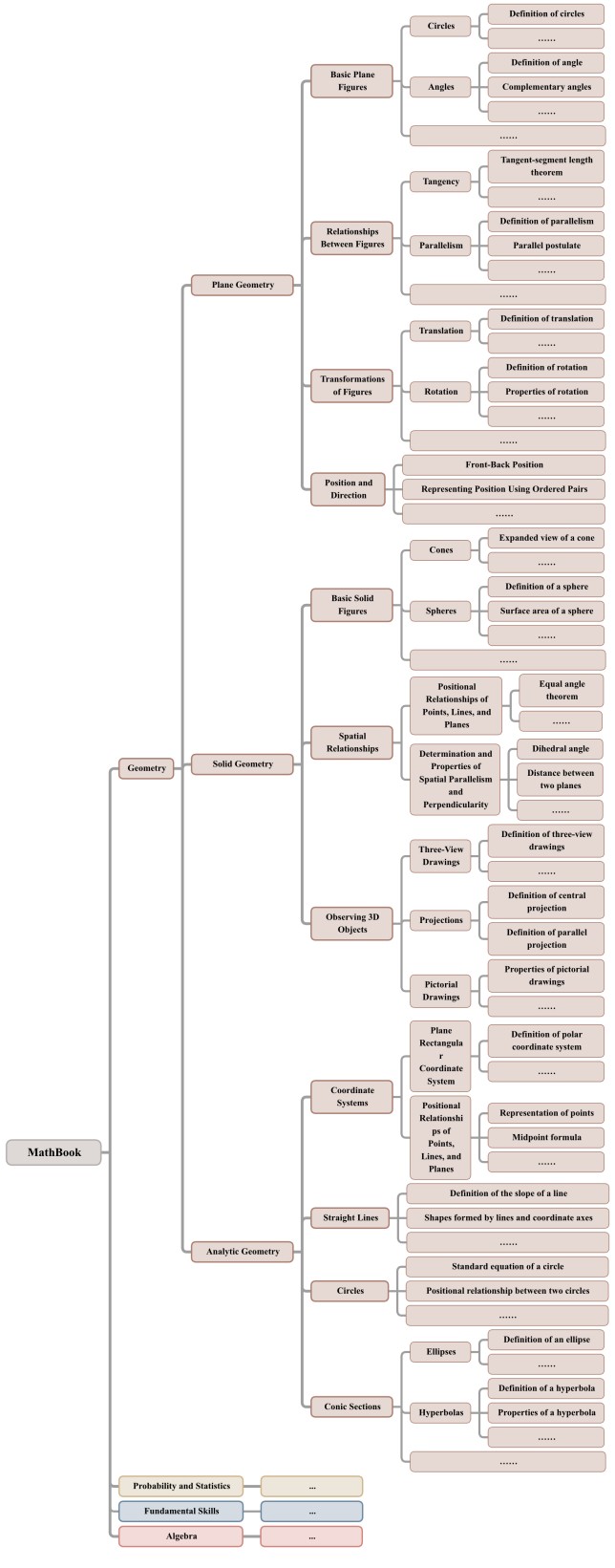

Figure 4: Overview of the hierarchical structure of knowledge points in MathBook (1).

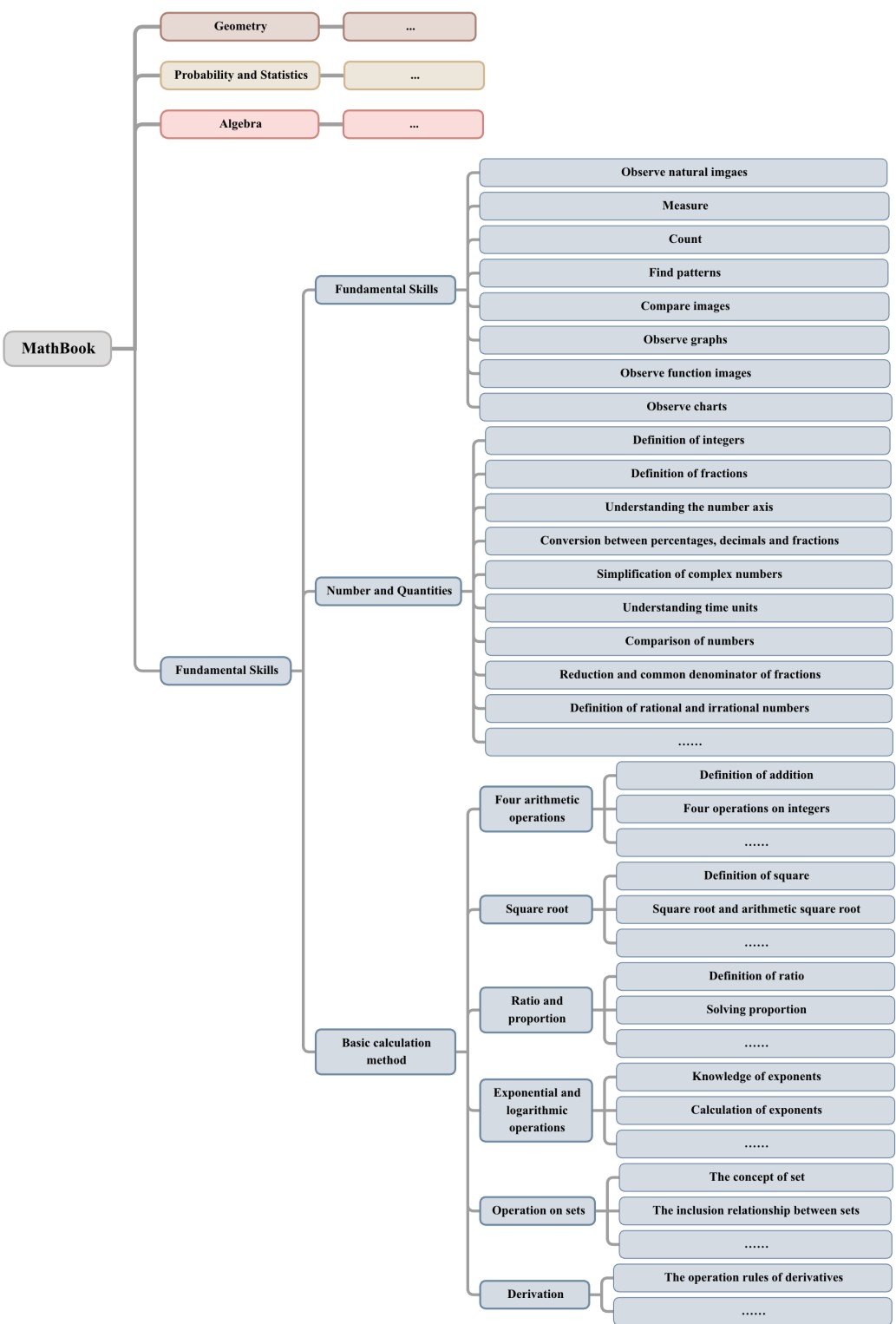

Figure 5: Overview of the hierarchical structure of knowledge points in MathBook (2).

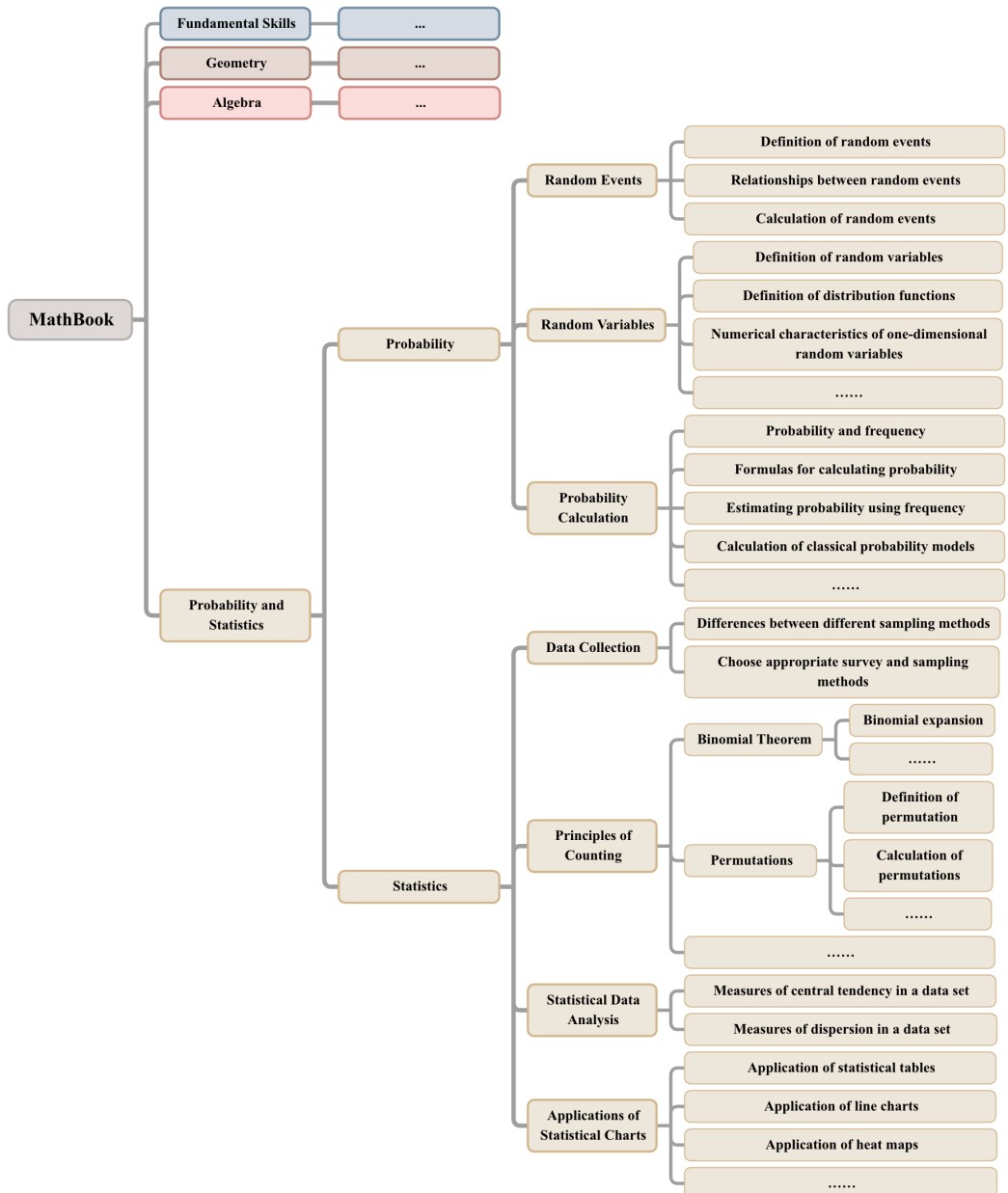

Figure 6: Overview of the hierarchical structure of knowledge points in MathBook (3).

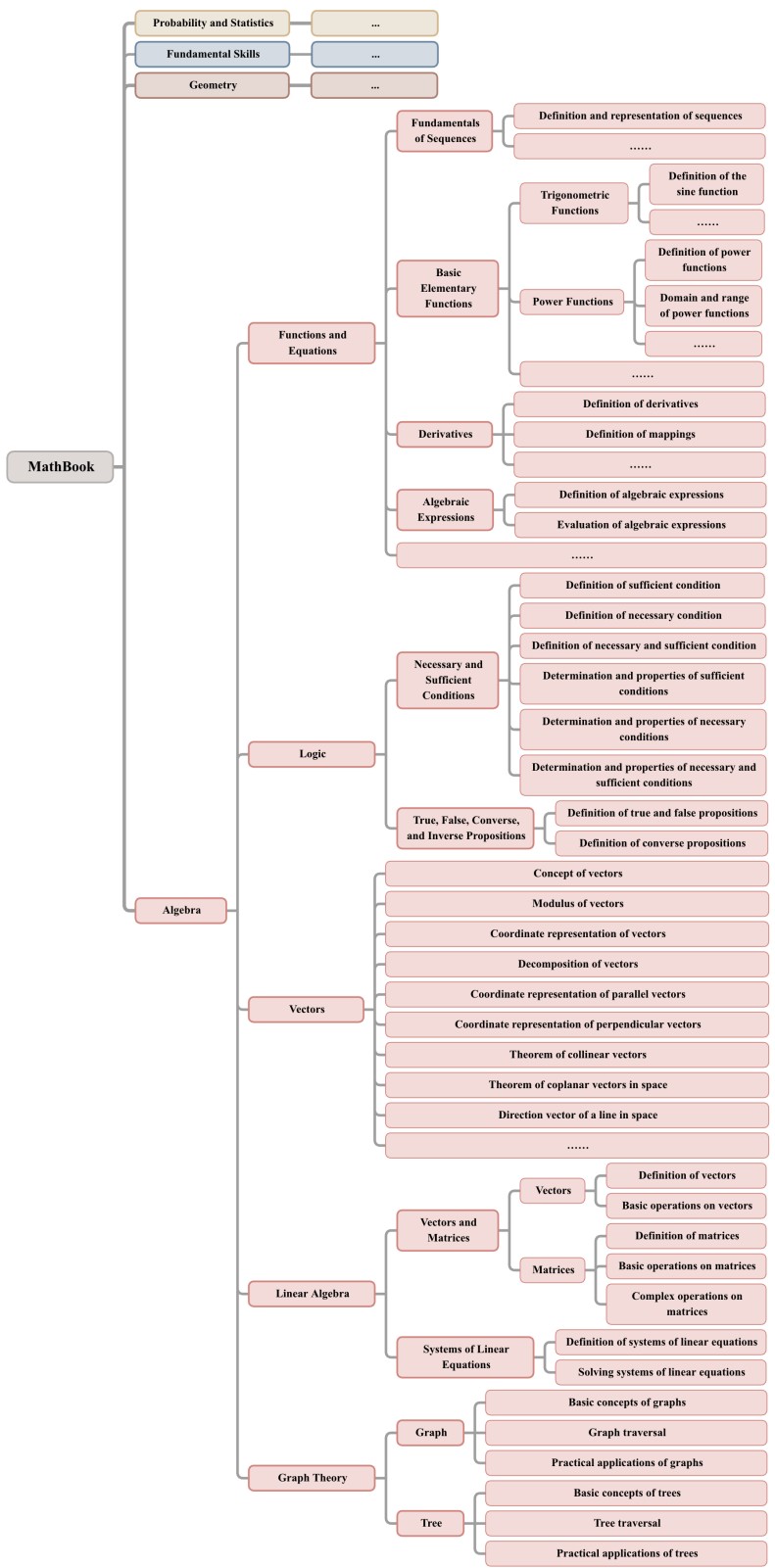

Figure 7: Overview of the hierarchical structure of knowledge points in MathBook (4).

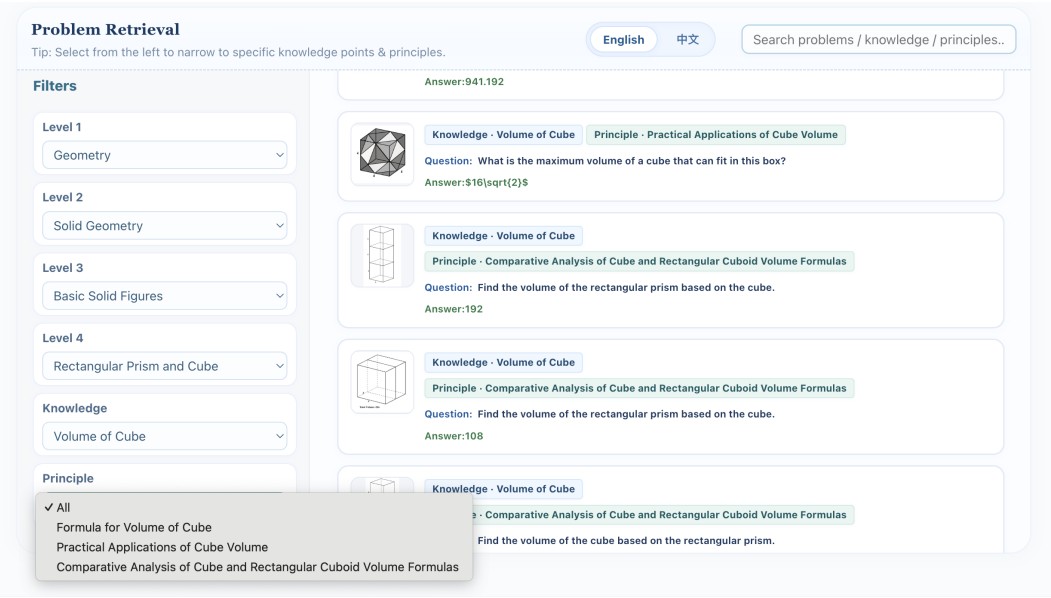

Figure 8: An example from We-Math 2.0 showing how each problem is explicitly aligned with a specific knowledge point and its associated principle.

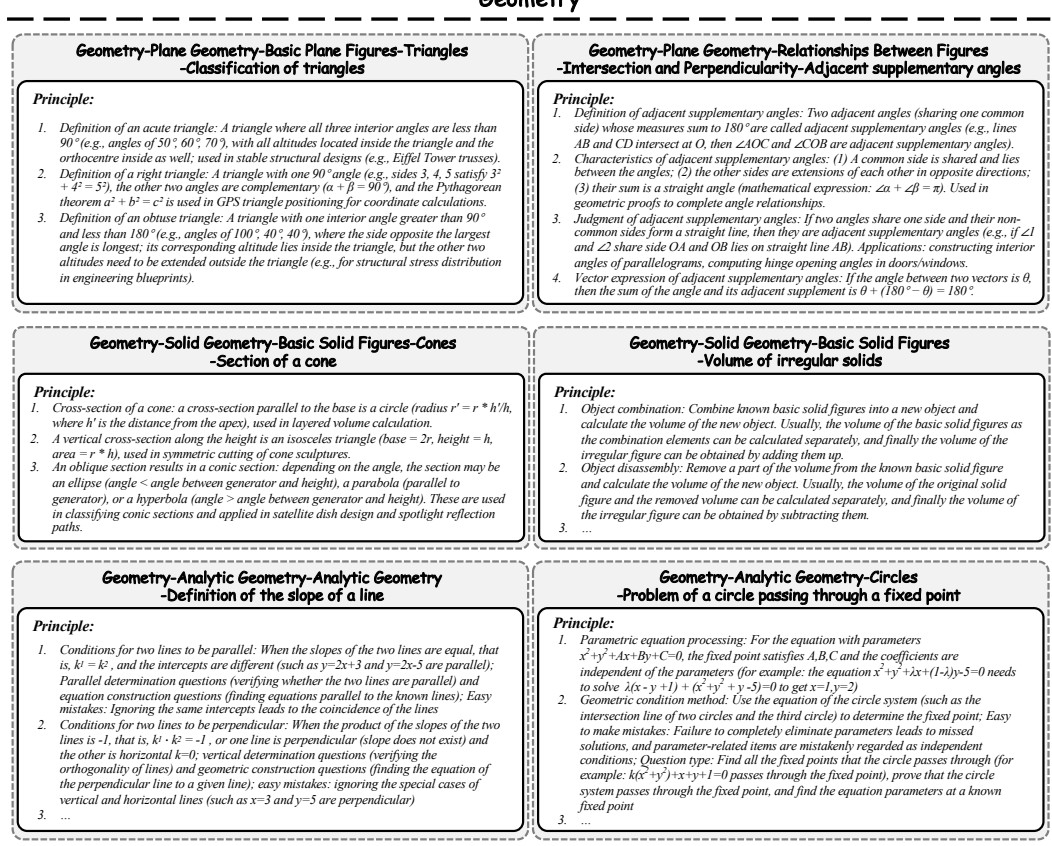

Figure 9: Examples of knowledge principles corresponding to specific knowledge points in MathBook (1).

## Fundamental Skills

**Fundamental Skills-Compare sizes in images**

*Principle:*

1. *Comparison of function values: Compare y-values of different functions at the same x-value.*
2. *Comparison of rate of change: Compare slopes (rate of increase/decrease) over the same interval.*
3. *Comparison of extrema: Compare maximum/minimum values of different functions.*
4. *Comparison of definite integrals: Compare the area enclosed between the curve and the x-axis.*

**Fundamental Skills-Understanding Numbers and Quantities-Compare sizes in images-Definition of integers**

*Principle:*

1. *Definition of natural numbers: Non-negative integers, used for counting or ordering.*
2. *Definition of positive integers, negative integers, and zero: Positive integers: Greater than 0, typically used for quantity. Zero: Neither positive nor negative, represents "none" or a reference point. Negative integers: Less than 0, representing opposite quantities.*
3. *Set notation: The set of integers is denoted by Z. The set of natural numbers is denoted by N.*

**Fundamental Skills-Basic Calculation Methods-Squaring and Square Rooting-Definition of square**

*Principle:*

1. *Definition of square: Square refers to the result of multiplying a number by itself, that is, a number multiplied by itself. For example, the square of 2 is $2 \times 2 = 4$. In mathematics, the symbol "²" is used to represent a square, for example, $2^2 = 4$*
2. *Non-negativity of square: The square of any real number is non-negative, that is, the square root does not have a negative solution (such as $(-3)^2 = 9 \geq 0$)*
3. *Parity of square: If a number is even, then its square is also an even number (such as $4^2 = 16$); if it is odd, its square is also an odd number (such as $5^2 = 25$)*
4. *Definition of square number: A square number is the square of an integer, which can be written in the form of a perfect square (such as $1 = 1^2$, $4 = 2^2$, $9 = 3^2$, $16 = 4^2$)*

**Fundamental Skills-Basic Calculation Methods-Four Arithmetic Operations-Definition of division**

*Principle:*

1. *The essential definition of division: the inverse operation of multiplication, that is, to find another factor given a product and a factor. If $a \times b = c$, then $c \div b = a$ (where $b \neq 0$), represented by "÷" or fraction line "/", such as $a \div b$ or $a/b$, the divisor cannot be zero (undefined in mathematics) does not satisfy the commutative law ($6 \div 2 \neq 2 \div 6$)*
2. *The concept of remainder: when it cannot be divided evenly, the remaining part is called the remainder (such as $7 \div 3 = 2$ remainder 1)*
3. *Practical application: allocate resources , calculate the average. Easy to make mistakes: the remainder unit is not marked, ignore the divisibility condition*
4. *Reciprocal association: $a \div b = a \times (1/b)$, division is converted into multiplication by the reciprocal*

Figure 10: Examples of knowledge principles corresponding to specific knowledge points in Math-Book (2).

## Probability and Statistics

**Probability and Statistics-Probability-Random Events -Definition of random events**

*Principle:*

1. *Random event: An experiment satisfies the condition that it can be repeated under the same conditions, and all possible results are clearly known. If the results of each experiment are uncertain, then the experiment is a random experiment, such as shooting a basketball and flipping a coin. Results that may or may not appear in an experiment are called random events. Events that must occur in each experiment are called inevitable events, and events that must not occur are called impossible events, denoted as empty set.*
2. *Sample space: Each possible result of a random experiment is called a sample point, and the set of all sample points is called the sample space. Random events are always composed of several basic events.*
3. *...*

**Probability and Statistics-Probability-Probability Calculation -Estimating probability using frequency**

*Principle:*

1. *The principle of frequency estimation probability: approximate the true probability by the frequency of events in a large number of repeated experiments (the mathematical basis is the law of large numbers: when the number of experiments is $n \to \infty$, the frequency is $f_n(A) \to P(A)$*
2. *Application scenarios: weather forecast (historical rainfall frequency predicts the probability of precipitation tomorrow), gambling game winning rate calculation (long-term statistics of roulette)*
3. *Formula example: The frequency of the number 6 appearing in 600 dice tossing is 98 times, and the estimated probability is $\hat{P} = \frac{98}{600} \approx 0.163$*

**Probability and Statistics-Statistics-Principles of Counting-Addition and Multiplication Principles-Step-by-step multiplication counting principle**

*Principle:*

1. *To complete a task with multiple steps, multiply the number of ways for each step (formula: $N = m_1 * m_2 * ... * m_k$, e.g., 4-digit password, each digit has 10 choices → $10^4 = 10,000$). Key condition: steps must be ordered and independent (e.g., "choose shirt" then "choose pants").*
2. *Application scenarios: password combinations (letters + numbers), travel routes (3 paths from A to B, 2 from B to C → total = $3 \times 2 = 6$)*
3. *Special cases: repetition allowed (e.g., repeated license plate letters), partial restrictions (e.g., phone number cannot start with 0). Question types: restricted permutations (e.g., 3-digit numbers formed from 1–9 with no repeats)*

**Probability and Statistics-Statistics-Principles of Counting-Combinations -Combination-based counting problems**

*Principle:*

1. *At least/at most problems: "at least m" is total combinations minus combinations less than m (e.g., "at least 3 qualified items" = total − C(n,0) − C(n,1) − C(n,2), formula: $C(n , k)$at_least_$m = \sum_{i=m}^n C(n , i)$, e.g., probability of at least 2 defective items in 10 = $1 − (C(10,0) + C(10,1)) / 2^{10}$*
2. *Element distribution problems: same items to different people → use divider method (e.g., 10 identical balls to 3 people, each gets ≥1: C(9,2) = 36), allowing empty box: C(n+k−1,k−1) (e.g., 7 balls into 4 boxes: C(10,3) = 120)*
3. *Balls into boxes problem: distinguishable balls into distinguishable boxes (m choices per ball → $m^n$); indistinguishable balls into distinguishable boxes (divider method); indistinguishable balls into indistinguishable boxes (integer partitions, e.g., Stirling numbers)*

Figure 11: Examples of knowledge principles corresponding to specific knowledge points in Math-Book (3).

## Algebra

---

### Algebra-Functions and Equations-Understanding Functions -Analytical expression of functions

**Principle:**
1. *Concept of analytical expression: A way to express a function using mathematical formula (e.g., uniform motion $s = vt$, direct proportionality $y = kx$); may be explicit (e.g., $y = x^3$) or implicit (e.g., $x^2 + y^2 = 1$).*
2. *Types of expressions: Linear (e.g., $y = 3x-2$), quadratic (e.g., $y = x^2+2x+1$), exponential (e.g., $y = 2^x$), logarithmic (e.g., $y = \ln x$), trigonometric (e.g., $y = A \sin(\omega x+\varphi)$), piecewise (e.g., $y = \{x, x \geq 0; -x, x < 0\}$).*
3. *Determining the expression: Given type, solve for parameters: use method of undetermined coefficients (e.g., let $y = kx + b$, plug in (2,5) and (3,7) → solve for k = 2, b = 1); given graph, derive expression using vertex (e.g., parabola with vertex (1,−3) → $y = a(x−1)^2−3$, then plug in a point to solve a), or asymptotes (e.g., hyperbola $y = k/x$ → determine k); construct expression from conditions (e.g., profit model $y = -x^2 + 50x − 100$, or physical law $y = A \cos(\omega t)$).*

### Algebra-Functions and Equations-Fundamentals of Sequences -Definition and representation of sequences

**Principle:**
1. *Definition of a sequence: A sequence is a collection of numbers arranged in a specific pattern, where each number is called a term, denoted as $a^1$, $a^2$, $a^3$... (e.g., the natural number sequence 1, 2, 3, ...)*
2. *Expression of terms: $a^1$ is the first term (initial term), $a^n$ is the nth term (e.g., in the sequence 2, 4, 6, ..., $a^1 = 2$, $a^2 = 4$)*
3. *Methods of representing a sequence: Listing method: directly listing finite terms (e.g., 1, 3, 5, 7, 9) or infinite terms (e.g., 1, 1/2, 1/3, ...); General term formula: using $a^n = f(n)$ (e.g., $a^n = 2n - 1$ generates the odd number sequence 1, 3, 5, ...); Recursive formula: defining a term based on previous terms (e.g., Fibonacci sequence $a^n = a^{[n-1]} + a^{[n-2]}$ with initial values $a^1 = 1$, $a^2 = 1$)*

### Algebra-Logic-Necessary and Sufficient Conditions -Definition of true and false propositions

**Principle:**
1. *True propositions: Statements consistent with facts or logic (e.g., "In an isosceles triangle, the base angles are equal" – true, "2+2=4" – true, "The Earth is a planet" – true); False propositions: Statements that contradict facts or logic (e.g., "1 > 2" – false, "The Sun revolves around the Earth" – false, "All prime numbers are odd" – false because 2 is an even prime)*
2. *Truth of conditional propositions: In statements like "If A then B" $(A \rightarrow B)$, it is false only when A is true and B is false; in all other cases, it is true (e.g., "If it rains, the ground is wet" is false when it rains but the ground is dry)*
3. *Compound propositions: Formed by logical connectors like and, or, not (e.g., "2+2=4 and 3>5" is false, "x>1 or x<0" is true when x=2)*
4. *Role of counterexamples: A single counterexample disproves a universal statement (e.g., "All birds can fly" is disproved by the ostrich)*

### Algebra-Vectors-Concept of vectors

**Principle:**
1. *Definition of vector: a quantity that has both magnitude (modulus) and direction (such as velocity, force), as opposed to scalars (such as temperature, mass). Two elements of a vector: magnitude (such as velocity rate) and direction (such as the direction of force).*
2. *Representation of vector: geometrically represented by directed line segments (such as the vector from the starting point (A) to the end point (B) is denoted by $\overline{AB}$, algebraically represented by coordinates (such as two-dimensional vector ($\boldsymbol{\alpha} = (3,4)$). Symbols of vectors: bold lowercase letters (such as $\boldsymbol{\alpha}$) or symbols with arrows (such as $\vec{a}$), zero vector is denoted by $\vec{a}$) (direction is arbitrary, modulus is 0)*
3. *Basics of vector operations: addition (parallelogram law or triangle law), subtraction (adding opposite vectors), scalar multiplication (changing modulus, direction may be reversed)*

### Algebra-Linear Algebra-Vectors and Matrices-Matrices -Definition of matrices

**Principle:**
1. *In linear algebra, a matrix is a rectangular array of numbers, which can be real numbers or complex numbers. Matrices are usually denoted by capital letters, such as A. Each number in a matrix is called an element or entry of the matrix.*
2. *The size of a matrix is determined by its number of rows and columns, typically expressed as m×n, where m is the number of rows and n is the number of columns. An m×n matrix is a rectangular array consisting of m rows and n columns, with a number at each position. A matrix can be expressed as $A=(a_{ij})m×n$, where $a_{ij}$ is an element of the matrix, i represents the row index, and j represents the column index.*
3. *...*

### Algebra- Graph Theory-Tree -Basic concepts of trees

**Principle:**
1. *Definition of a tree: A tree is an undirected graph that is connected. There are no cycles in a tree. When a tree has n vertices, it has exactly (n-1) edges.*
2. *Basic terms for a tree: Root, the starting vertex of the tree, has no parent vertex. Parent node, the direct predecessor of a vertex. Child node, the direct successor of a vertex. Sibling node, a vertex with the same parent node. Ancestor, all vertices on the path from the root to the vertex. Descendants, all vertices from the vertex to the leaves. Leaves, vertices with no child nodes. Internal nodes, non-leaf nodes. Depth, the length of the path from the root to a vertex. Height, the length of the path from a vertex to the farthest leaf.*
3. *...*

Figure 12: Examples of knowledge principles corresponding to specific knowledge points in Math-Book (4).

### B.2    MathBook-Standard

#### B.2.1    GeoGebra-based Diagram Generation.

As described in the main text, all diagrams in MathBook-Standard are rendered using **GeoGebra**, a dynamic mathematics software that enables precise and reproducible geometric constructions. GeoGebra supports both interactive design and programmatic generation of diagrams, making it highly suitable for large-scale dataset creation. In our workflow, each problem is paired with a high-quality diagram constructed in GeoGebra, ensuring mathematical rigor and visual clarity. For automated and scalable generation, we leverage GeoGebra's ability to encode diagrams as scripts (e.g., XML, as shown in Figure 13), which allows for efficient parameter variation and reproducibility, but the core advantage lies in GeoGebra's expressive power and accuracy for mathematical figures.

Compared to general-purpose plotting libraries such as Python's `matplotlib`, GeoGebra offers richer geometric primitives and more precise control over mathematical relationships, supporting a broader range of problem types and visual styles (Table 6). As shown in Figures 14, 15, and 16, diagrams generated by GeoGebra exhibit higher fidelity and better alignment with mathematical conventions, which is essential for both algorithmic evaluation and educational use. It is evident that the complexity and precision of these diagrams would be difficult to achieve using Python-based plotting tools alone.

Table 6: Comparison between GeoGebra and Python Plotting tools.

| Tool | Command Line Plotting | Interactive Graphic Editing | Image Re-editing | Precise Parameter Control |
|------|-----------------------|-----------------------------|------------------|---------------------------|
| GeoGebra | ✓ | ✓ | ✓ | ✓ |
| Python Plotting | ✓ | ✗ | ✗ | Limited |

#### B.2.2    Dataset Diversity and Variant Construction.

Building on the GeoGebra-based pipeline, we systematically construct a diverse dataset as detailed in the main text. For each knowledge point and principle, a **seed problem** is designed with a corresponding diagram. To further enhance diversity, we introduce two types of variants: **one-problem-multi-image** (generating multiple diagram instances for the same problem by varying parameters in GeoGebra) and **one-image-multi-problem** (curating multiple questions for a single diagram, each derived from different knowledge points or mathematical principles). Representative examples of seed problems and their variants are shown in Figure 17– 24, demonstrating the flexibility and extensibility of our approach.

By leveraging GeoGebra's capabilities, MathBook-Standard achieves both high-fidelity geometric representation and systematic dataset expansion, ensuring rich semantic and visual diversity for mathematical reasoning tasks.

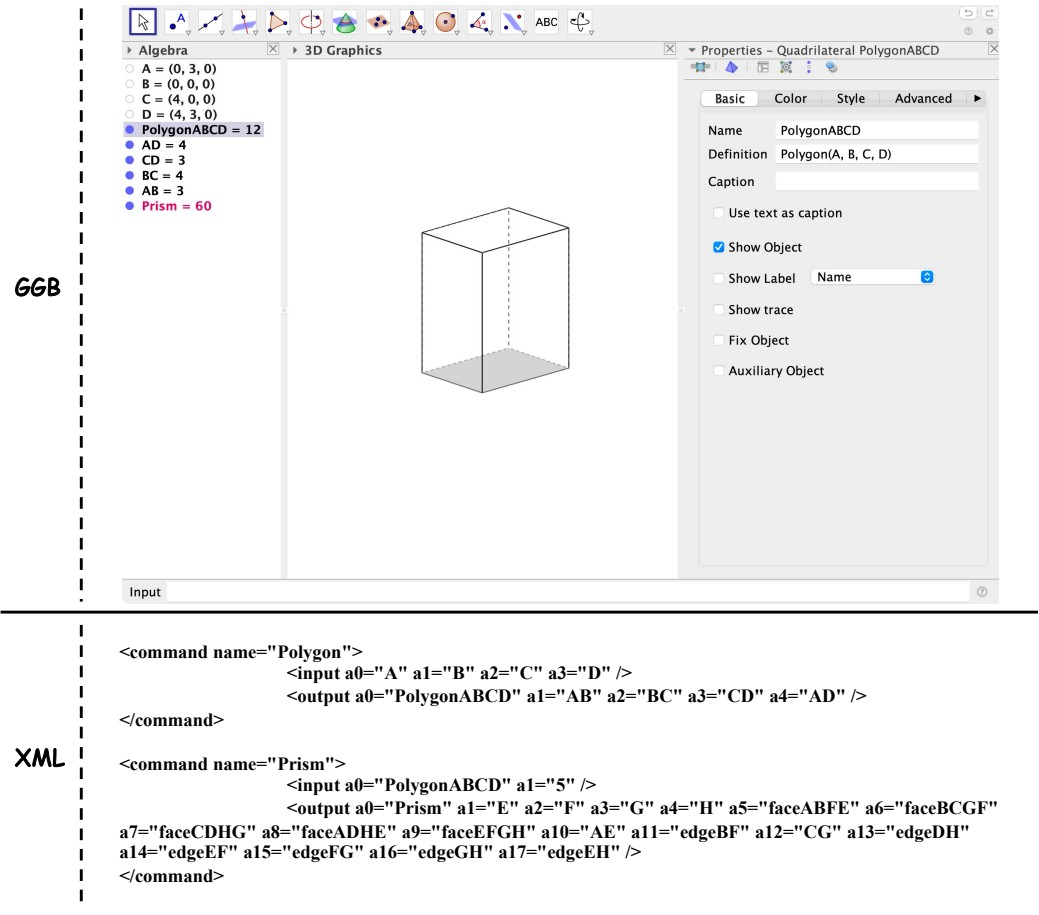

Figure 13: Overview of the GeoGebra interface and part of the corresponding XML script, showing core commands for defining geometric objects.

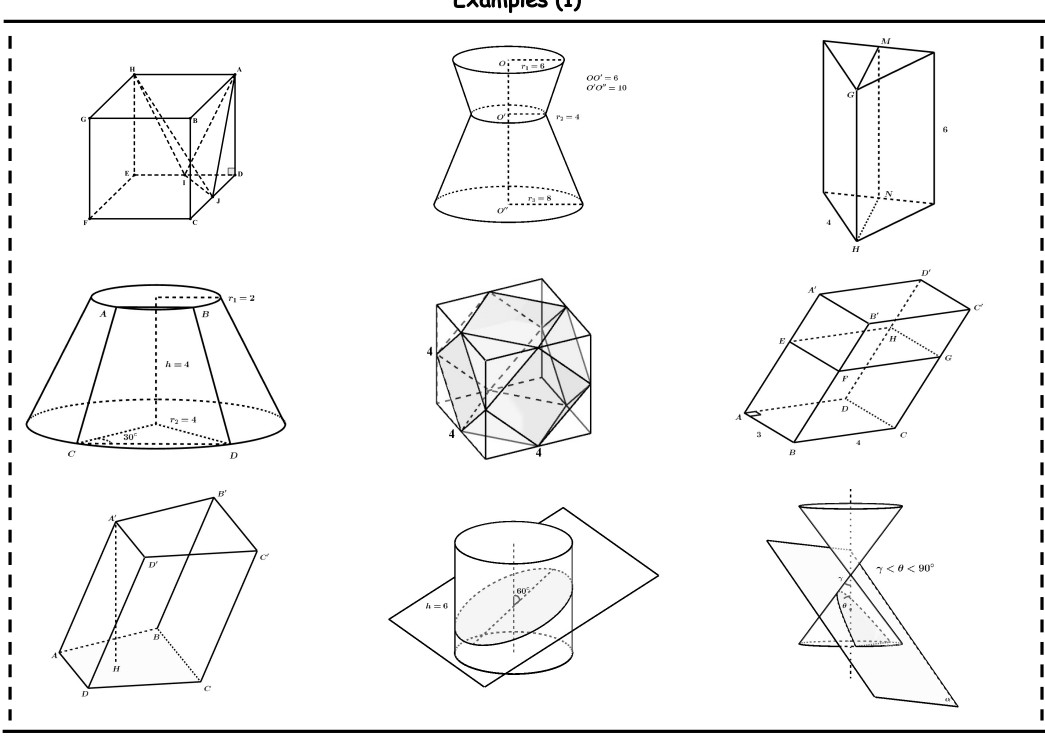

Figure 14: Overview of a group of GeoGebra-generated images in MathBook (1).

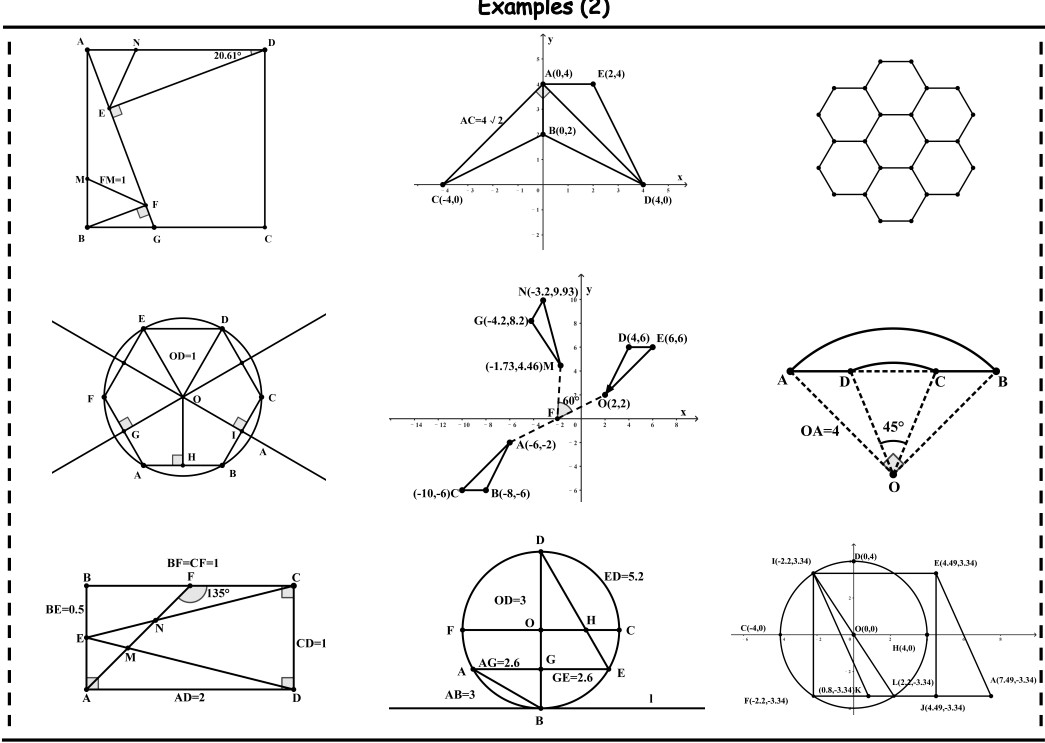

Figure 15: Overview of a group of GeoGebra-generated images in MathBook (2).

**Examples (3)**

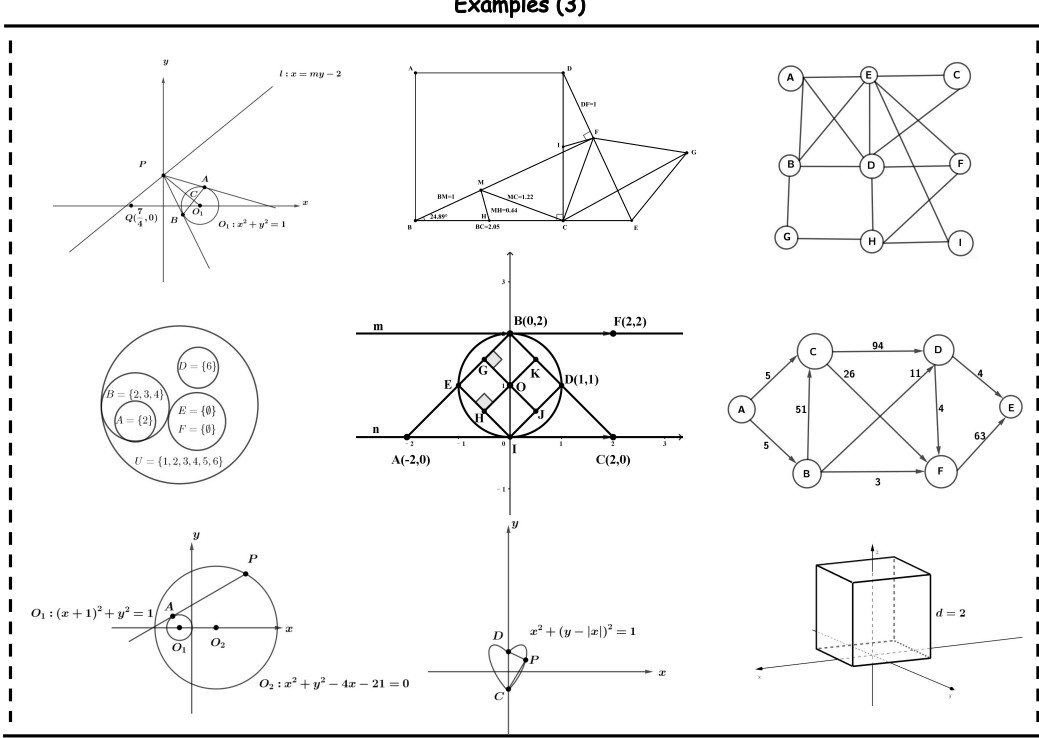

Figure 16: Overview of a group of GeoGebra-generated images in MathBook (3).

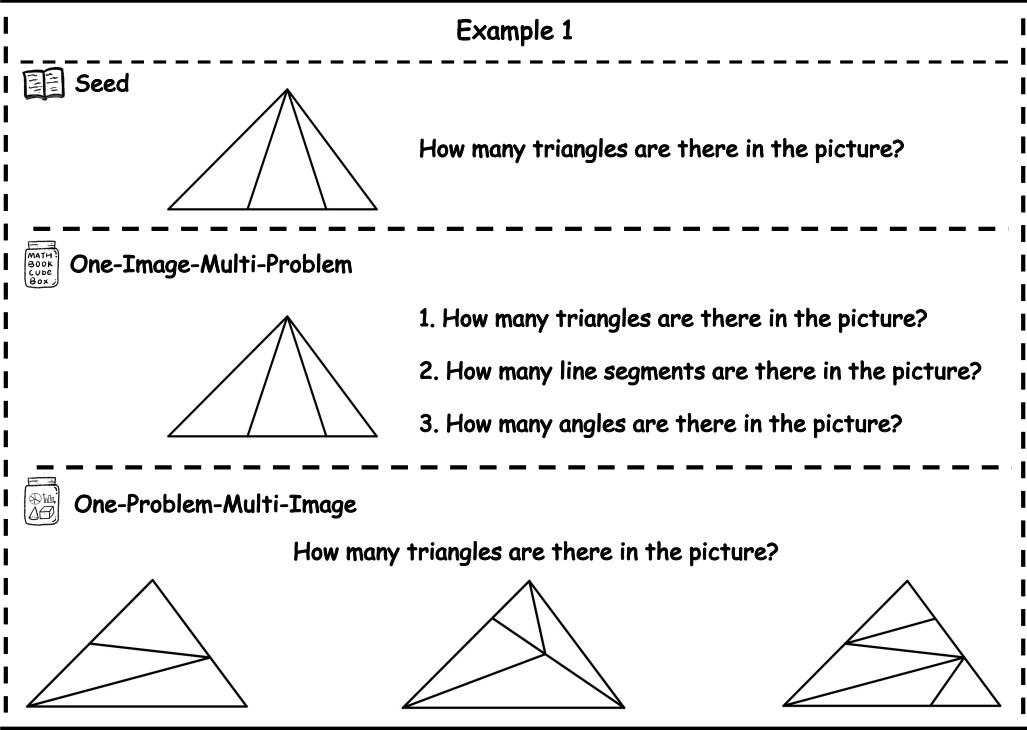

Figure 17: An example of MathBook-Standard data instance (1).

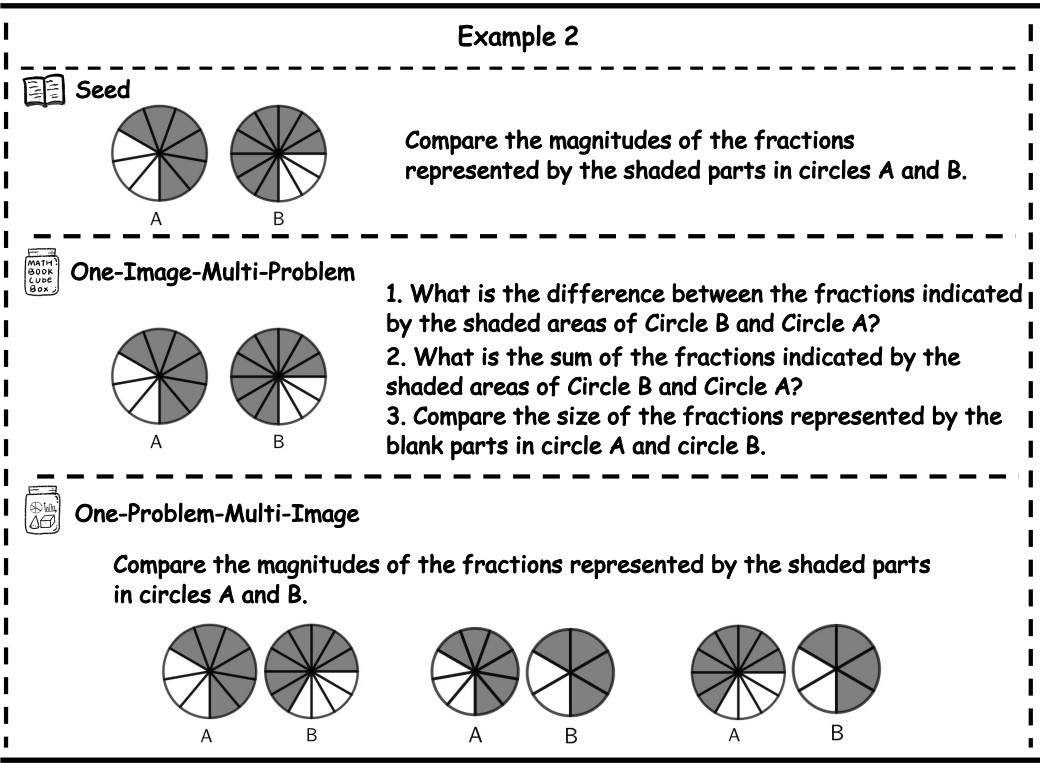

Figure 18: An example of MathBook-Standard data instance (2).

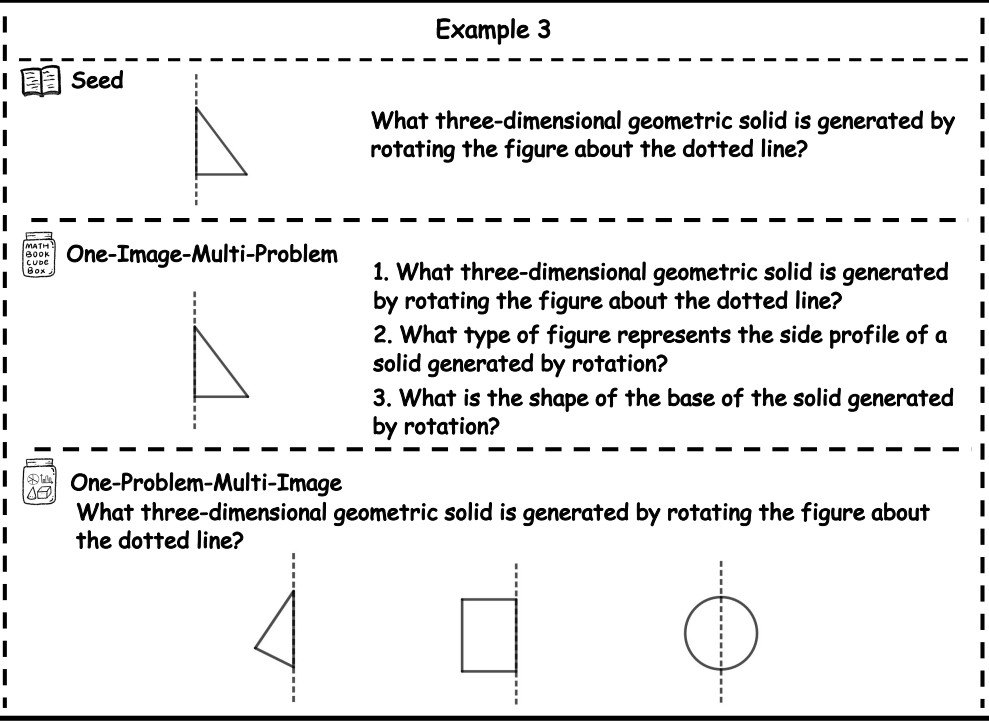

Figure 19: An example of MathBook-Standard data instance (3).

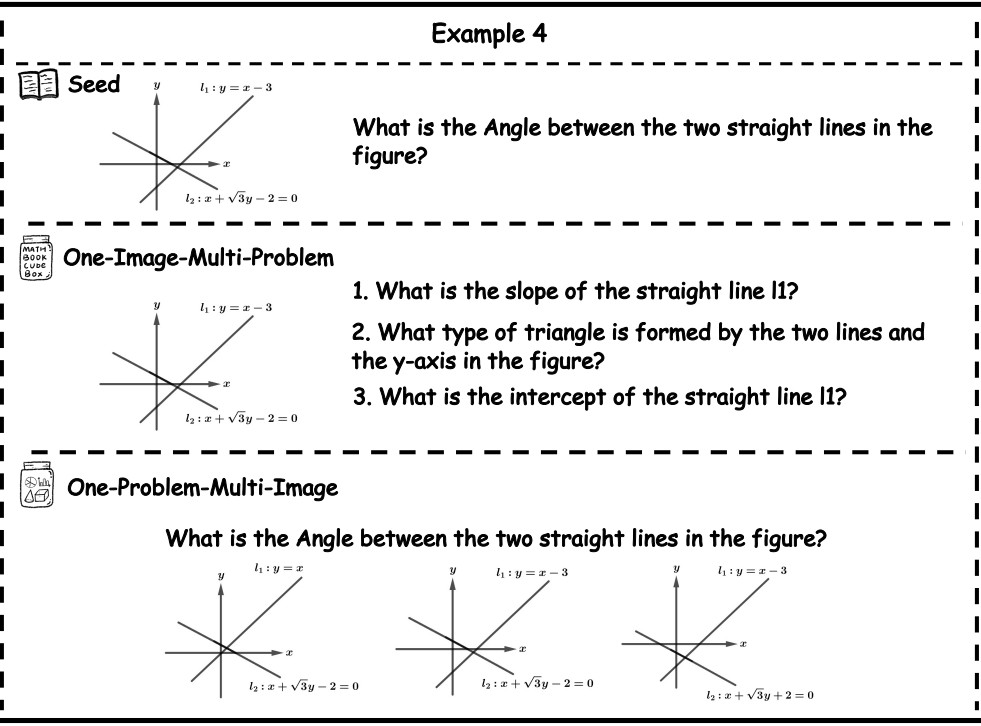

Figure 20: An example of MathBook-Standard data instance (4).

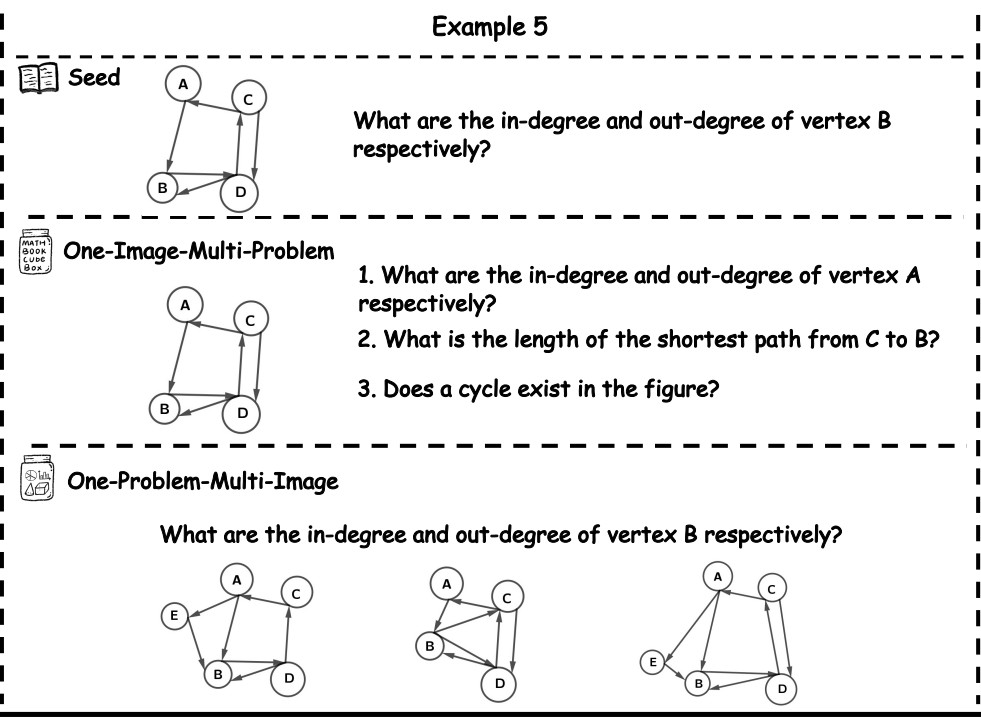

Figure 21: An example of MathBook-Standard data instance (5).

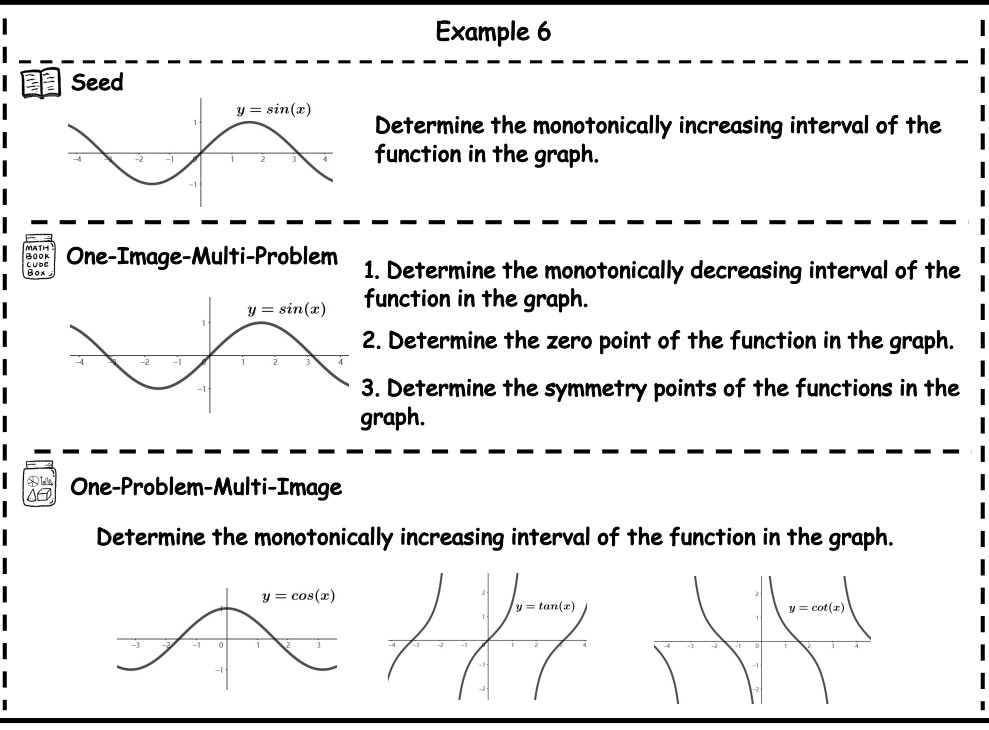

Figure 22: An example of MathBook-Standard data instance (6).

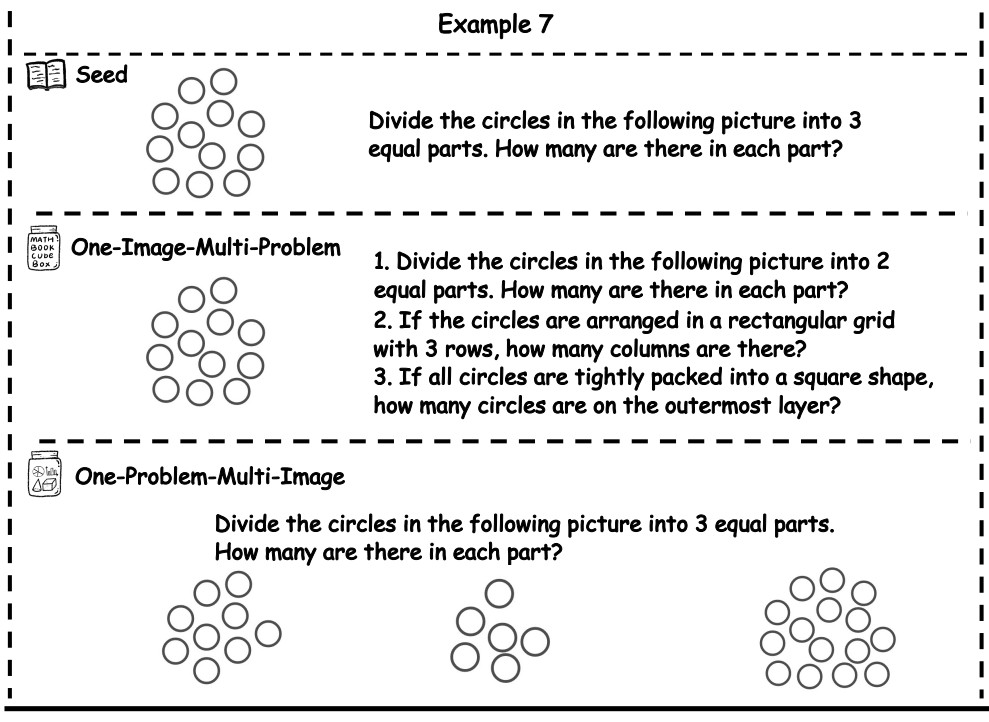

Figure 23: An example of MathBook-Standard data instance (7).

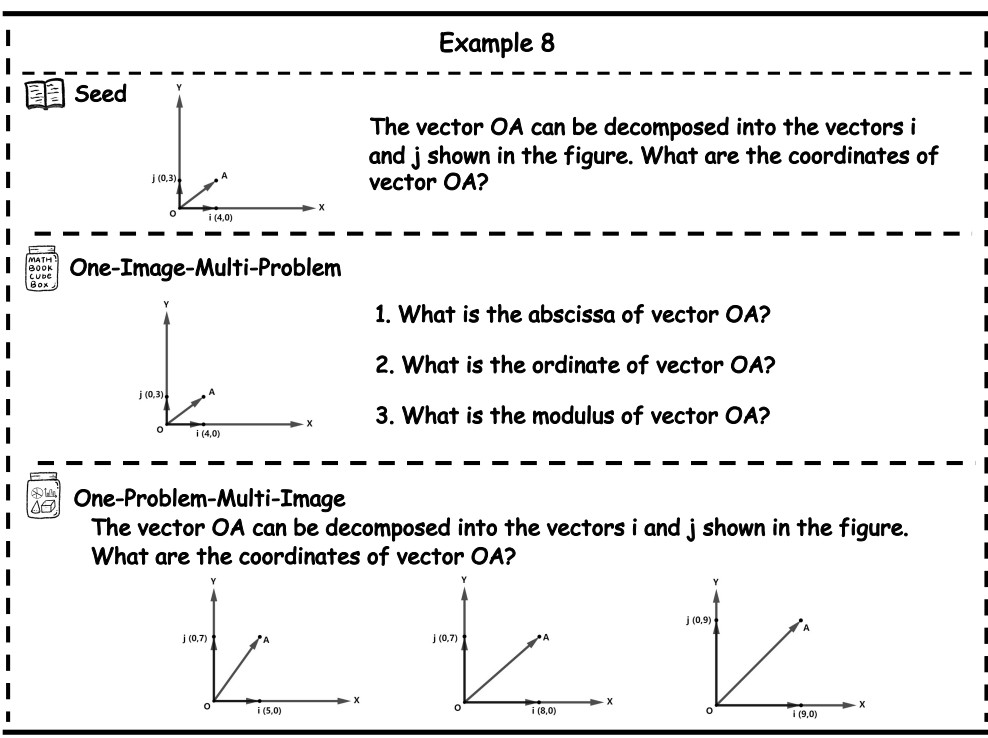

Figure 24: An example of MathBook-Standard data instance (8).

B.3 MATHBOOK-PRO

As shown in Figure 25, we present a concrete example from MathBook-Pro to illustrate the construction and expansion of problem variants within the three-dimensional difficulty space. The seed problem, positioned at the origin, focuses on the *arc length formula* in plane geometry, and involves knowledge points such as *definition of angle*, *definition of circles*, *four arithmetic operations of integers* and *four arithmetic operations of fractions*.

We first demonstrate how the seed problem is expanded along each individual dimension:

**Step Complexity:** The number of required knowledge points is increased by introducing new intermediate conclusions as conditions. In this example, the extended variant requires not only the *arc length formula*, but also incorporates *circumference of a circle* and *area of a circle* as additional knowledge points. The solution to the new problem depends on the answer to the seed problem, reflecting a progressive deepening of reasoning.

**Visual Complexity:** The original diagram is enhanced by adding shaded regions, which increases the visual and interpretive demands while keeping the core mathematical focus unchanged.

**Contextual Complexity:** The problem statement is recontextualized from a direct geometric description to a real-world application scenario. Although the narrative becomes more complex, the essential assessment of the *arc length formula* remains at the core.

By systematically combining these single-dimension expansions, we further generate multi-dimensional variants that integrate increased step, visual, and contextual complexity. In total, starting from the seed problem, we construct 7 variants corresponding to all possible combinations of the three dimensions. Each new variant is constructed through progressive modifications to both the problem statement and the accompanying image, resulting in a diverse and interpretable set of difficulty-controlled problems. The full set of variants and their corresponding dimensions are summarized in Table 7.

As shown in Figure 26, MathBook-Pro supports the Dynamic Scheduling RL framework by providing difficulty-controlled problem variants. These structured training samples enable progressive adaptation across different difficulty levels in a dynamic training process.

Table 7: Difficulty-controlled variants constructed from the seed problem in MathBook-Pro. Each variant corresponds to a unique combination of step, visual, and contextual complexity.

| Variant | Step | Visual | Contextual |
|---|---|---|---|
| Seed | - | - | - |
| Variant 1 | ✓ | - | - |
| Variant 2 | - | ✓ | - |
| Variant 3 | - | - | ✓ |
| Variant 4 | ✓ | ✓ | - |
| Variant 5 | ✓ | - | ✓ |
| Variant 6 | - | ✓ | ✓ |
| Variant 7 | ✓ | ✓ | ✓ |

B.4 MATHBOOKEVAL

B.4.1 DATASET CONSTRUCTION AND ANNOTATION PROTOCOL.

To ensure both comprehensive knowledge coverage and rigorous, interpretable annotations for visual mathematical reasoning, MathBookEval is constructed through a multi-stage, process-oriented pipeline. We begin by integrating representative samples collected from five open-source benchmarks: MathVista (Lu et al., 2023), MathVerse (Zhang et al., 2024a), MathVision (Wang et al., 2024), We-Math (Qiao et al., 2024a) and DynaMath (Zou et al., 2024), systematically filtering out redundant or highly similar items to maximize diversity in knowledge point combinations and reasoning patterns. All problems are re-annotated under unified guidelines, ensuring consistency in annotation style and

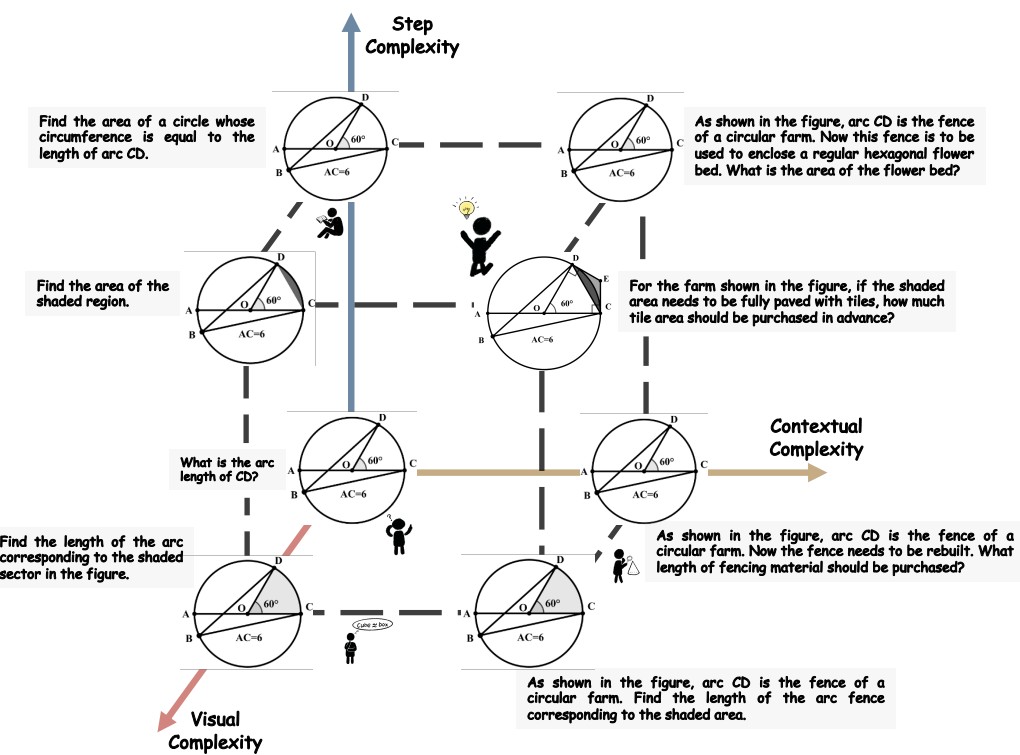

Figure 25: An example from MathBook-Pro in the difficulty space.

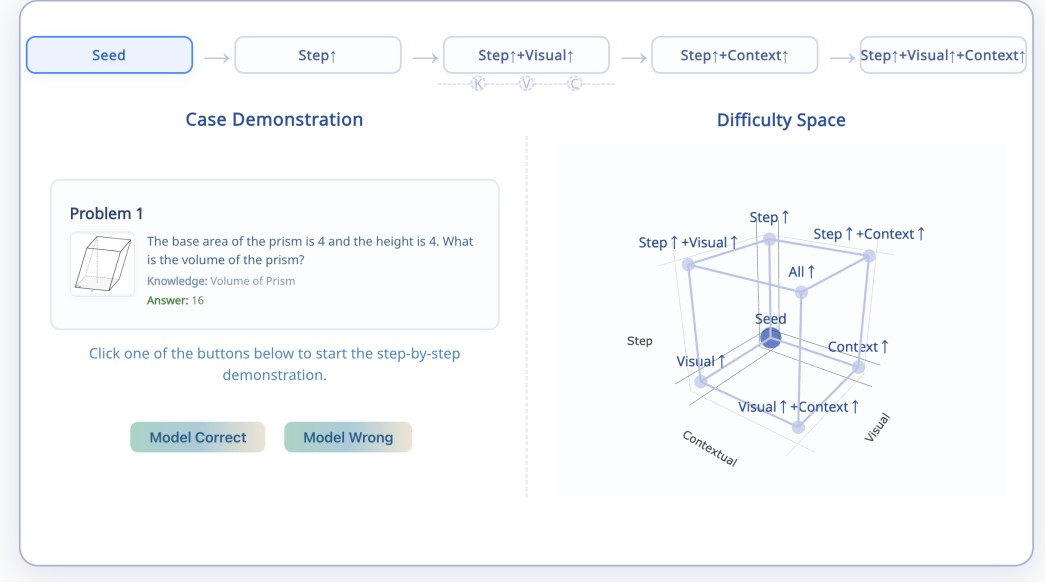

Figure 26: Overview of the Dynamic Scheduling RL training stage with MathBook-Pro.

granularity. For each problem, at least two human experts independently provide a complete, step-by-step solution, where each step is explicitly decomposed according to the underlying knowledge point(s) from the unified knowledge system $\mathcal{K}$. This knowledge-point-based decomposition is fundamental: it enables precise and systematic quantification of reasoning depth, as each reasoning

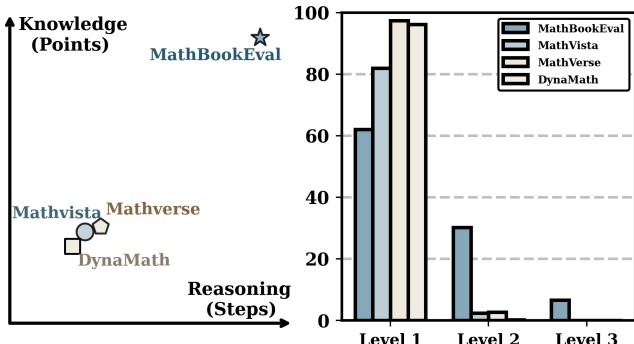

Figure 27: Comparison of MathBookEval and open-source benchmarks

step directly corresponds to a specific knowledge point. Only those problems for which the set of annotated knowledge points at each step is exactly consistent across expert annotations are retained in the final dataset, ensuring high reliability and objectivity. To address knowledge points and reasoning depths insufficiently covered by existing benchmarks, additional samples are newly constructed by human experts following the same process-oriented, knowledge-point-level annotation protocol. As a result, MathBookEval comprises both collected samples from open-source benchmarks and newly constructed samples, achieving balanced and comprehensive coverage across mathematical domains and reasoning complexities, with every annotation step tightly aligned to the relevant knowledge points for robust reasoning depth modeling.

### B.4.2 TASK DIMENSIONS.

MathBookEval is organized along two dimensions, capturing both the diversity of mathematical knowledge and the depth of reasoning required for problem solving.

**(1) Reasoning Dimension:** Problems are categorized by the number of reasoning steps required to reach the solution. Each step is explicitly defined and directly mapped to a specific knowledge point in the unified knowledge system $\mathcal{K}$. We define three levels: Level 1 (1–3 steps, basic reasoning), Level 2 (4–6 steps, intermediate reasoning), and Level 3 (7–10 steps, complex reasoning). This step-by-step annotation based on knowledge points allows for clear and objective quantification of reasoning depth, enabling detailed analysis of model performance across different levels of reasoning complexity.

**(2) Knowledge Dimension:** The 491 knowledge points in $\mathcal{K}$ are distributed across 4 top-level domains and 13 subdomains, all of which are covered in the benchmark. Each problem is annotated with all the knowledge points involved in its solution, and every reasoning step is aligned with a specific knowledge point. This enables fine-grained evaluation of model capabilities across various mathematical topics and educational stages.

### B.4.3 DATASET STATISTICS.

Table 8 summarizes the key statistics of MathBookEval. The benchmark contains 1,000 fully annotated problems, covering all 491 knowledge points in the unified knowledge system. Of these, 600 are sourced from open-source benchmarks and 400 are newly constructed to address coverage gaps and increase diversity. Problems are distributed across multiple formats, including multiple-choice and fill-in-the-blank, and span a wide range of reasoning depths and knowledge domains.

As shown in Figure 27, we compare MathBookEval and existing benchmarks. In the right panel, the y-axis represents the percentage of problems at each reasoning level. Note that for some benchmarks, the total does not reach 100% because we exclude problems that experts annotate as belonging to other subjects such as physics, chemistry, or biology, in order to ensure a rigorous comparison. It is evident from the figure that existing benchmarks contain less than 3% of problems at Level 2 (4–6 steps), and none at Level 3 (7–10 steps). In contrast, MathBookEval substantially supplements these two categories, providing a more comprehensive evaluation of multi-step reasoning abilities.

Table 8: Key statistics of MathBookEval.

| Statistics | Number |
|---|---|
| Total Problems | 1,000 |
| - Open-source Benchmarks | 600 |
| - MathVista | 150 |
| - MathVerse | 150 |
| - MathVision | 100 |
| - We-Math | 100 |
| - DynaMath | 100 |
| - Newly Constructed | 400 |
| Knowledge Points Covered | 491 |
| - Domains | 4 |
| - Subdomains | 13 |
| Reasoning Depth | |
| - Level 1 (1–3 steps) | 62.0% |
| - Level 2 (4–6 steps) | 30.2% |
| - Level 3 (7–10 steps) | 7.8% |

### B.4.4 EVALUATION PROTOCOL AND METRICS.

MathBookEval incorporates existing evaluation protocols, including both LLM-as-a-judge and rule-based approaches. To ensure consistency, MathBookEval adopts the LLM-as-a-judge protocol as the evaluation rule. Specifically, we adopt the LLM-as-a-judge protocol, following MathVista and MathVerse by employing GPT-4o as the judge model and reporting overall accuracy as the evaluation metric. The detailed prompt used for evaluation is shown in Table 9.

Table 9: Prompt templates for evaluation on MathBookEval.

| Type | Prompt Template |
|---|---|
| Evaluation Prompt | Now, we require you to solve a math question. Please briefly describe your thought process and provide the final answer.
For multiple-choice questions, return the selected option and its content. For direct answer selection, return only the chosen result. For fill-in-the-blank questions, answer directly.
**Question**: \<Question\>
Regarding the format, please answer following the template below, and be sure to include two \<\> symbols:
**\<Thought process\>**: \<\<your thought process\>\>
**\<Answer\>**: \<\<your answer\>\> |

### B.4.5 EXPERIMENT SETUP

**Details of the Evaluated Models.** To evaluate the performance of various multimodal large models on mathematical tasks, we include a diverse set of recent models in our benchmark. Table 10 lists the release dates and official sources of all evaluated models. Additionally, Table 11 provides an overview of their architectural designs to support a comprehensive comparison.

**Details of the Model Hyperparameters.** For all closed-source models accessed via API, we adopt the standard generation settings and perform inference on CPUs, with the process typically completing within a day. For open-source models, inference is conducted on a cluster equipped with 8 NVIDIA A800-SXM4-80GB GPUs, using the hyperparameter configurations provided in the official inference examples. If no specific instructions are available, default settings are applied. The detailed generation parameters are summarized in Table 12 and Table 13.

Table 10: The release time and model source of MLLMs used in MathBookEval

| Model | Release Time | Source |
|---|---|---|
| GPT-4o (OpenAI, 2024) | 2024-05 | `https://gpt4o.ai/` |
| GPT-4V (OpenAI, 2023) | 2024-04 | `https://openai.com/index/gpt-4v-system-card/` |
| InternVL2.5-78B (Chen et al., 2024b) | 2024-12 | `https://huggingface.co/OpenGVLab/InternVL2_5-78B` |
| InternVL2.5-8B (Chen et al., 2024b) | 2024-12 | `https://huggingface.co/OpenGVLab/InternVL2_5-8B` |
| Qwen2.5-VL-72B (Bai et al., 2025a) | 2025-01 | `https://huggingface.co/Qwen/Qwen2.5-VL-72B-Instruct` |
| Qwen2.5-VL-7B (Bai et al., 2025a) | 2025-01 | `https://huggingface.co/Qwen/Qwen2.5-VL-7B-Instruct` |
| Qwen2.5-VL-3B (Bai et al., 2025a) | 2025-01 | `https://huggingface.co/Qwen/Qwen2.5-VL-3B-Instruct` |
| LLaVA-OneVision-72B (Li et al., 2024) | 2024-08 | `https://huggingface.co/lmms-lab/llava-onevision-qwen2-72b-ov-chat` |
| LLaVA-OneVision-7B (Li et al., 2024) | 2024-08 | `https://huggingface.co/lmms-lab/llava-onevision-qwen2-7b-ov` |
| GLM-4V-9B (GLM et al., 2024) | 2024-06 | `https://huggingface.co/THUDM/glm-4v-9b` |

Table 11: Model architecture of 10 MLLMs evaluated on MathBookEval.

| Models | LLM | Vision Encoder |
|---|---|---|
| GPT-4o | - | - |
| GPT-4V | - | - |
| InternVL2.5-78B | Qwen2.5-72B-Instruct | InternViT-6B-448px-V2_5 |
| InternVL2.5-8B | internlm2_5-7b-chat | InternViT-6B-448px-V2_5 |
| Qwen2.5-VL-72B | Qwen2.5-72B-Instruct | CLIP ViT-bigG-P14 |
| Qwen2.5-VL-7B | Qwen2.5-7B-Instruct | CLIP ViT-bigG-P14 |
| Qwen2.5-VL-3B | Qwen2.5-3B-Instruct | CLIP ViT-bigG-P14 |
| LLaVA-OneVision-72B | Qwen2-72B | SigLip-so400m-P14-384 |
| LLaVA-OneVision-7B | Qwen2-7B | SigLip-so400m-P14-384 |
| GLM-4V-9B | GLM-9B | EVA_02_CLIP-E-P14 |

### B.4.6 ADDITIONAL RESULTS ON MATHBOOKEVAL

Table 14 reports the complete results of all evaluated models on MathBookEval, providing a more comprehensive perspective on model behavior across both reasoning depth and knowledge-coverage dimensions. Overall, general-purpose open-source MLLMs exhibit stable performance trends, with larger models such as Qwen2.5-VL-72B and InternVL2.5-78B demonstrating strong accuracy across multiple reasoning levels and knowledge categories. In contrast, reasoning-oriented systems (for example, MM-Eureka-7B) show performance patterns influenced by their reasoning-focused training pipelines, excelling in certain multi-step inference settings while presenting varied outcomes across different knowledge domains. These results collectively illustrate how different training paradigms and objectives shape mathematical reasoning capabilities in complementary ways. Within this broader landscape, MathBook-7B achieves competitive performance across both reasoning and knowledge dimensions, reflecting the effectiveness of the structured knowledge design and progressive alignment strategies introduced in We-Math 2.0.

Furthermore, as shown in Figure 31 to Figure 40, we present the performance of different models on various subdomains in MathBookEval, where subdomains belonging to the same domain are indicated with the same color. It can be observed that models generally perform worse on geometry-related problems, especially in subdomains such as solid geometry and analytic geometry, which involve higher visual complexity and require more advanced reasoning. In contrast, models tend to achieve better results on algebra and fundamental skills, particularly in subdomains related to computational methods.

Table 12: Generating parameters for Open-Source MLLMs.

| Model | Generation Setup |
|---|---|
| InternVL2.5-78B | do_sample = False, max_new_tokens = 1024 |
| InternVL2.5-8B | do_sample = False, max_new_tokens = 1024 |
| Qwen2.5-VL-72B | do_sample = False, max_new_tokens = 1024 |
| Qwen2.5-VL-7B | do_sample = False, max_new_tokens = 1024 |
| Qwen2.5-VL-3B | do_sample = False, max_new_tokens = 1024 |
| LLaVA-OneVision-72B | do_sample = False, max_new_tokens = 1024 |
| LLaVA-OneVision-7B | do_sample = False, max_new_tokens = 1024 |
| GLM-4V-9B | do_sample = False |

Table 13: Generating parameters for Closed-Source MLLMs.

| Model | Generation Setup |
|---|---|
| GPT-4o | "model" : "gpt-4o", "temperature" : 0, "max_tokens" : 1024 |
| GPT-4V | "model" : "gpt-4-turbo", "temperature" : 0, "max_tokens" : 1024 |

Table 14: Full results of different MLLMs on MathBookEval. **Acc.**: Accuracy; **FS.**: Foundational skills; **PS.**: Probability and statistics; **Geo.**: Geometry; **Alg.**: Algebra.

| Models | Acc. | Reasoning | | | Knowledge | | | |
|---|---|---|---|---|---|---|---|---|
| | | Level1 | Level2 | Level3 | FS. | PS. | Geo. | Alg. |
| *Closed-source MLLMs* | | | | | | | | |
| GPT-4o | 50.8 | 52.8 | 48.9 | 41.7 | 33.8 | 57.6 | 44.2 | 67.2 |
| GPT-4V | 42.8 | 44.0 | 43.0 | 31.9 | 36.8 | 56.6 | 33.5 | 59.4 |
| *Open-source (General)* | | | | | | | | |
| InternVL2.5-78B | 51.8 | 52.5 | 51.8 | 45.8 | 50.0 | 64.2 | 42.6 | 67.6 |
| Qwen2.5-VL-72B | 57.1 | 58.3 | 56.4 | 50.0 | 52.9 | 58.5 | 52.1 | 68.8 |
| LLaVA-OneVision-72B | 43.0 | 44.8 | 42.0 | 31.9 | 38.2 | 52.8 | 37.0 | 53.5 |
| InternVL2.5-8B | 37.9 | 40.7 | 34.5 | 27.8 | 33.8 | 46.2 | 31.4 | 50.0 |
| Qwen2.5-VL-7B | 46.7 | 50.1 | 43.0 | 33.3 | 44.1 | 58.5 | 38.8 | 60.2 |
| LLaVA-OneVision-7B | 31.6 | 34.3 | 28.0 | 23.6 | 36.8 | 41.5 | 24.9 | 41.0 |
| Qwen2.5-VL-3B | 36.9 | 38.7 | 34.2 | 33.3 | 35.3 | 49.1 | 29.1 | 49.6 |
| GLM-4V-9B | 22.2 | 23.7 | 20.5 | 16.7 | 26.5 | 23.6 | 18.4 | 28.9 |
| *Open-source (Reasoning)* | | | | | | | | |
| MM-Eureka-7B | 50.0 | 50.6 | 51.5 | 38.9 | 47.1 | 52.8 | 44.0 | 62.9 |
| R1-VL-7B | 38.0 | 41.9 | 32.6 | 27.8 | 38.2 | 38.7 | 32.3 | 50.4 |
| OpenVLThinker-7B | 32.6 | 33.0 | 32.6 | 29.2 | 45.6 | 38.7 | 24.9 | 43.4 |
| VLAA-Thinker-7B | 35.7 | 38.3 | 31.9 | 29.2 | 46.6 | 42.5 | 30.2 | 41.6 |
| **MathBook-7B** | 50.4 | 52.0 | 48.2 | 45.8 | 57.4 | 67.9 | 40.5 | 63.3 |

## C  MORE DETAILS ON MATHBOOK-RL

### C.1  IMPLEMENTATION DETAILS

We use Qwen2.5-VL-7B-Instruct as the base model and conduct all experiments on 8×A800 GPUs. The training process consists of two stages.

In the first stage, we perform cold-start supervised fine-tuning (SFT) to help the model develop explicit awareness of knowledge system and a knowledge-driven reasoning paradigm. The SFT stage uses a learning rate of $1.0 \times 10^{-5}$, is trained for 1 epoch, and adopts a warmup ratio of 0.1.

In the second stage, we apply dynamic reinforcement learning (RL) to further improve the model's generalization ability. For RL, we set the rollout temperature to 1.0 and generate 8 rollouts per sample. The learning rate is set to $1 \times 10^{-6}$, and the maximum completion length is 1024 tokens. The system prompt used in this stage is illustrated in Table 15. The reward function combines answer accuracy (weight 0.9) and response format compliance (weight 0.1). Specifically, we use MathVerify to extract and compare answers for accuracy, while format compliance ensures the output follows the required structure.

Table 15: The system prompt template for response generation in the RL stage.

| Type | Prompt Template |
| --- | --- |
| System Prompt | A conversation between User and Assistant. The user asks a question, and the Assistant solves it. The assistant first outputs the thinking process in <think> </think> and then provides the final answer(number, option, phrase, or LaTeX expression as appropriate) in <answer> </answer> tags. User: {question} Assistant: |

## C.2 ADDITIONAL EXPERIMENTAL RESULTS FOR MATHBOOK-3B

Table 16 summarizes the performance of MathBook-3B across multiple benchmarks. Our analysis yields the following observations.

Table 16: Performance comparison between Qwen2.5-VL-3B and MathBook-3B across benchmarks.

| Model | MathVista | MathVision | WeMath | MathVerse | MathBookEval |
| --- | --- | --- | --- | --- | --- |
| Qwen2.5-VL-3B | 60.8 | 21.1 | 22.9 | 29.1 | 36.9 |
| MathBook-3B | 63.2 | 24.3 | 28.1 | 32.6 | 40.2 |

**(1) Consistent improvements across benchmarks.** MathBook-3B surpasses the Qwen2.5-VL-3B backbone on all evaluated datasets, with notable gains on WeMath (+5.2), MathBookEval (+3.3), and MathVerse (+3.5). These results indicate that the structured data design and training methodology introduced in We-Math 2.0 remain effective even at smaller model scales, enhancing both multimodal mathematical reasoning and generalization to diverse problem formats.

**(2) Performance gains independent of model size.** Despite using a significantly smaller backbone, MathBook-3B achieves improvements comparable in trend to those observed with MathBook-7B. This suggests that the benefits of We-Math 2.0 arise from its principled knowledge structure and training pipeline, rather than relying on large model capacity.

## C.3 GENERALIZATION TO TEXT-ONLY MATHEMATICAL REASONING TASKS

To further examine whether the structured knowledge design in We-Math 2.0 transfers beyond the multimodal setting, we additionally evaluate MathBook-7B on two widely used text-only mathematical reasoning benchmarks: GSM8K (Cobbe et al., 2021) and MATH-500 (Lightman et al., 2023). As shown in Table 17, MathBook-7B achieves consistent improvements over the Qwen2.5-VL-7B backbone. Although these datasets differ substantially from the multimodal training distribution, the gains observed here suggest that the structured knowledge supervision introduced in We-Math 2.0 can support more generalizable reasoning behavior across modalities.

## C.4 CASE STUDY

To further illustrate the strengths of our approach, we present several representative case studies comparing our model with the Qwen2.5-VL-7B baseline across different benchmarks.

Table 17: Results on text-only reasoning benchmarks.

| Model | MATH-500 | GSM8K |
|---|---|---|
| Qwen2.5-VL-7B | 66.0 | 86.2 |
| MathBook-7B | 67.2 | 87.6 |

**Case 1: Conciseness and Reasoning Process.** Figure 28 compares the response patterns of our model and the Qwen2.5-VL-7B baseline on We-Math. Our model produces more concise answers, with a reduced average response length, while still retaining all necessary formulas and reasoning steps. This demonstrates that our approach effectively mitigates the issue of overthinking, resulting in more focused and efficient solutions without sacrificing completeness or mathematical rigor.

**Case 2: Spatial Reasoning Enhancement.** Figure 29 highlights the performance of our model on MathVision. Compared to the baseline, which often fails to correctly interpret or solve such problems, our model (MathBook-7B) shows clear improvements in spatial reasoning. This is particularly evident in questions requiring the understanding of geometric relationships or positional logic, indicating that our training strategy significantly enhances the model's ability to handle spatially complex scenarios.

**Case 3: Knowledge-Oriented and Context-Aware Mathematical Reasoning.** In Figure 30, we compare the responses of our model and the Qwen2.5-VL-7B baseline on MathVista. Our model not only applies the relevant mathematical concepts correctly but also demonstrates improved integration of mathematical knowledge with real-world problem contexts. This case exemplifies the model's strengthened ability to bridge abstract mathematical reasoning and practical application, reflecting the benefits of our knowledge-oriented and context-aware training paradigm.

Overall, these case studies provide qualitative evidence that our model achieves more concise, accurate, and contextually appropriate reasoning compared to strong baselines, particularly in scenarios demanding spatial understanding and knowledge-oriented problem solving.

### C.5 EXPERIMENT SETUP

#### C.5.1 DETAILS OF THE EVALUATION.

We evaluate our model on four representative benchmarks: MathVista (Lu et al., 2023), Math-Verse (Zhang et al., 2024a), MathVision (Wang et al., 2024), and We-Math (Qiao et al., 2024b). During evaluation, we strictly follow the official scoring protocols provided in the respective benchmark GitHub repositories to ensure fair and consistent comparison. Specifically, we report results on the testmini split for MathVista, MathVerse, MathVision, and We-Math. For We-Math, we adopt the main evaluation metric as defined in the original paper, reporting "Score (Strict)" as the primary metric. For the other benchmarks, we report the average accuracy as the main evaluation result.

#### C.5.2 DETAILS OF THE BASELINES

We compare our method with a comprehensive set of baselines from three perspectives: closed-source models (GPT-4o (OpenAI, 2024), Gemini-1.5-Pro (Team et al., 2023)), open-source general models (Qwen2.5-VL-7B (Bai et al., 2025a), InternVL2.5-8B (Chen et al., 2023)), GLM-4V-9B (GLM et al., 2024), LLaVA-OneVision-7B (Li et al., 2024), and open-source reasoning models (Math-PUMA-7B (Zhuang et al., 2024), URSA-8B (Luo et al., 2025), R1-Onevision-7B (Yang et al., 2025b), R1-VL-7B (Zhang et al., 2025), MM-Eureka-7B (Meng et al., 2025), WeThink-7B (Yang et al., 2025a), VLAA-Thinker-7B (Chen et al., 2025), OpenVLThinker-7B (Deng et al., 2025)), which enables a thorough and multi-faceted comparison to highlight the advantages of our approach.

**GPT-4o (OpenAI, 2024):** GPT-4o ("o" for omni) is OpenAI's 2024 flagship multilingual, multimodal large language model that accepts text, images, and audio as input for unified cross-modal understanding. It is designed for broad adaptability and seamless integration of visual and linguistic information, supporting complex reasoning across modalities and languages.

**Gemini-1.5-Pro (Team et al., 2023):** Gemini 1.5 Pro, developed by Google DeepMind, is a multimodal model capable of processing text, images, audio, and video inputs, with an extremely

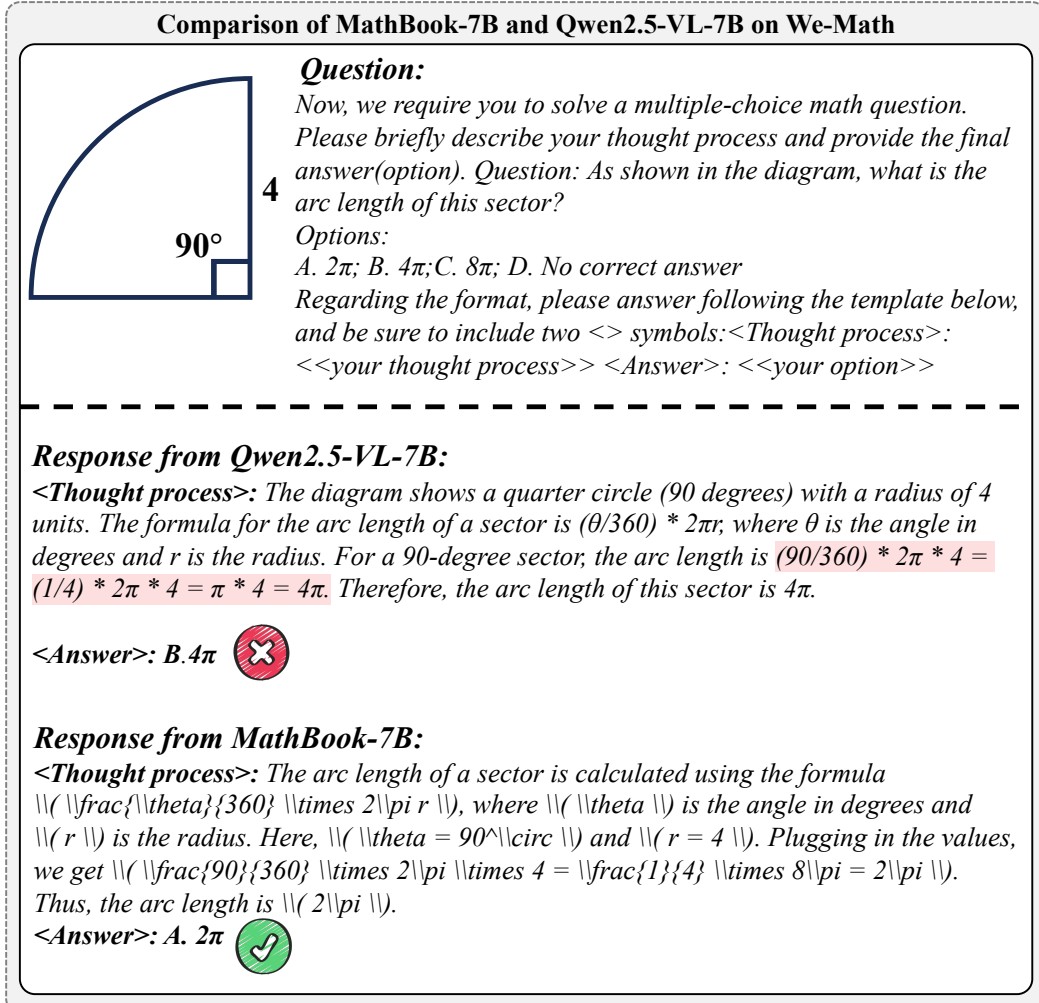

Figure 28: Comparison of MathBook-7B and Qwen2.5-VL-7B on We-Math.

long context window (up to 1–2 million tokens). It is optimized for multi-task proficiency and structured tool integration, enabling analysis of lengthy and diverse content.

**Qwen2.5-VL-7B (Bai et al., 2025a):** Qwen2.5-VL-7B is an open-source vision-language model with 7 billion parameters, designed to generate both free-form text and structured outputs such as bounding boxes and JSON. It emphasizes fine-grained visual understanding and multi-modal alignment, supporting tasks like document analysis and event detection.

**InternVL2.5-8B (Chen et al., 2023):** InternVL2.5-8B is an 8B-parameter open-source multimodal model that employs progressive scaling and co-training strategies to align its vision and language components. It incorporates training optimizations and a curated dataset to enhance cross-modal reasoning and reduce hallucinations.

**GLM-4V-9B (GLM et al., 2024):** GLM-4V-9B is a 9B-parameter multimodal model built on the GLM series architecture. It integrates visual understanding and language processing capabilities, and is optimized using large-scale multimodal data. With specialized architectural design and training strategies, it demonstrates strong performance in multimodal reasoning tasks and is capable of effectively handling a variety of vision-centric tasks.

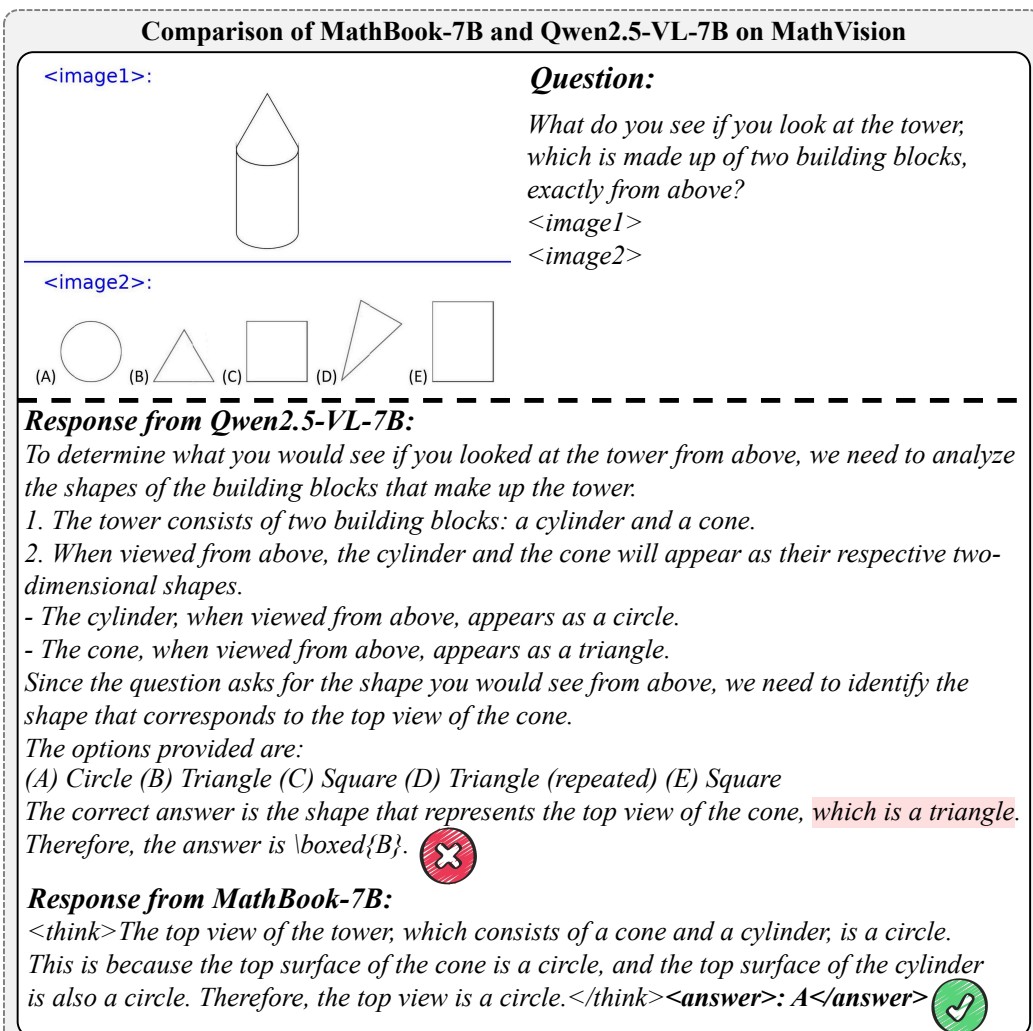

Figure 29: Comparison of MathBook-7B and Qwen2.5-VL-7B on MathVision.

**LLaVA-OneVision-7B (Li et al., 2024):** LLaVA-OneVision-7B is an open-source model designed for unified single-image, multi-image, and video understanding. It adopts an AnyRes visual representation strategy to enable cross-scenario capability transfer. Trained via a three-stage curriculum, it performs well across various benchmarks and exhibits emerging capabilities like video-to-video difference analysis and multi-camera self-driving video understanding via task transfer from images.

**Math-PUMA-7B (Zhuang et al., 2024):** Math-PUMA-7B is a vision-language model focused on mathematical reasoning with visual inputs, introducing a three-stage curriculum for aligning textual and visual modalities. The model is optimized for visual math benchmarks and aims to ensure consistent problem-solving across formats such as text and diagrams.

**URSA-8B (Luo et al., 2025):** URSA-8B is an 8B-parameter multimodal model targeting chain-of-thought reasoning in visual mathematical problems, trained on large-scale multimodal CoT datasets. It employs a reward model for stepwise verification, emphasizing both the generation and validation of reasoning chains for reliable solutions.

**R1-Onevision-7B (Yang et al., 2025b):** R1-Onevision-7B is a 7B-parameter multimodal reasoning model that converts images into structured textual representations for symbolic reasoning. It is trained

---

**Comparison of MathBook-7B and Qwen2.5-VL-7B on MathVista**

| Average waiting time at the DMV | |
|---|---|
| **Month** | **Waiting time (minutes)** |
| August | 17 |
| September | 14 |
| October | 26 |
| November | 17 |
| December | 25 |

*Question:*

*Please answer the question requiring an integer answer and provide the final value, e.g., 1, 2, 3, at the end.*

*Question: An administrator at the Department of Motor Vehicles (DMV) tracked the average wait time from month to month. According to the table, what was the rate of change between August and September? (Unit: minutes per month)*

- - - - - - - - - - - - - - - - - - - - - - - - - - - - - - - - -

*Response from Qwen2.5-VL-7B:*

*<Thought process>: To find the rate of change between August and September, we need to calculate the difference in waiting time and divide it by the number of months between August and September.*

*The waiting time in August is 17 minutes.*
*The waiting time in September is 14 minutes.*
*The difference in waiting time is:*
*17 minutes (August) - 14 minutes (September) = 3 minutes*
*Since there is only one month between August and September, the rate of change is:*
*3 minutes / 1 month = 3 minutes per month*
*Therefore, the rate of change between August and September is 3 minutes per month.* 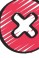

*Response from MathBook-7B:*

*<think>To find the rate of change between August and September, we calculate the difference in waiting times and divide by the number of months between them. The waiting time in August is 17 minutes, and in September it is 14 minutes. The change is \( 17 - 14 = 3 \) minutes. Since the time decreased, the rate of change is \(-3\) minutes per month.</think> <answer>-3</answer>*

Figure 30: Comparison of MathBook-7B and Qwen2.5-VL-7B on MathVista.

on step-by-step multimodal reasoning annotations and refined with reinforcement learning, enabling precise multi-hop visual-textual inference.

**R1-VL-7B (Zhang et al., 2025):** R1-VL-7B is an open-source 7B vision-language model designed for stepwise reasoning, applying Step-wise Group Relative Policy Optimization (StepGRPO) for dense intermediate rewards. This approach improves logical coherence and multi-step problem-solving, especially in mathematical and logical tasks.

**MM-Eureka-7B (Meng et al., 2025):** MM-Eureka-7B is a vision-language model based on Qwen2.5-VL-7B, fine-tuned with the MMK12 dataset and a rule-based reinforcement learning strategy. It is designed for multidisciplinary visual reasoning in math and science, using rule-based rewards to guide the learning of complex reasoning steps.

**WeThink-VL-7B (Yang et al., 2025a):** WeThink-VL-7B is a 7B-parameter general-purpose vision-language reasoning model fine-tuned on Qwen2.5-VL-7B via reinforcement learning. It is trained on the WeThink dataset and adopts a hybrid reward mechanism combining rule-based verification. The model enhances performance across both mathematical reasoning and general multimodal tasks by leveraging a scalable multimodal QA synthesis pipeline for diverse data generation and GRPO.

**OpenVLThinker-7B (Deng et al., 2025):** OpenVLThinker-7B is an open-source model tailored for complex vision-language reasoning, built by iterating between lightweight supervised fine-tuning and curriculum reinforcement learning. SFT initially distills chain-of-thought traces from text-based reasoning models, while RL refines these behaviors via a two-stage curriculum. It achieves improvements across six benchmarks, outperforming concurrent models with smaller training data.

**VLAA-Thinker-7B (Chen et al., 2025):** VLAA-Thinker-7B is an open-source large vision-language model optimized for multimodal reasoning, trained via Group Relative Policy Optimization with a novel mixed reward module. The mixed reward integrates 4 types of rule-based rewards and 1 open-ended reward, avoiding "pseudo reasoning paths" induced by supervised fine-tuning. It achieves good performance on the open LMM reasoning leaderboard.

# D  THE USE OF LARGE LANGUAGE MODELS (LLMS)

LLMs were used solely for grammar correction and minor language polishing. The conception, methodology, experiments, and analysis presented in this paper were entirely designed and implemented by the authors without relying on LLMs.

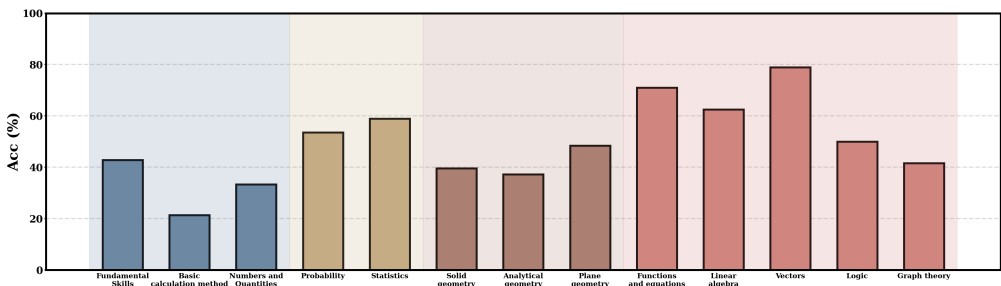

Figure 31: Detailed performance of GPT-4o across 13 subdomains.

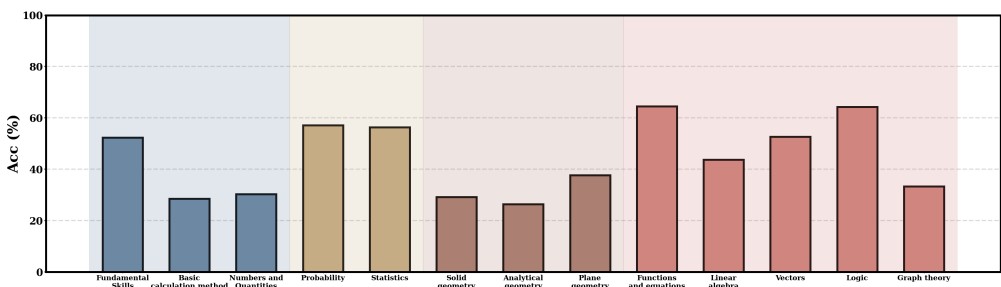

Figure 32: Detailed performance of GPT-4V across 13 subdomains.

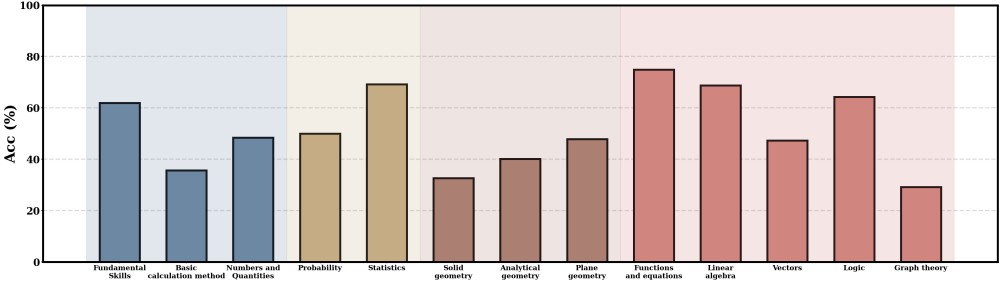

Figure 33: Detailed performance of InternVL2.5-78B across 13 subdomains.

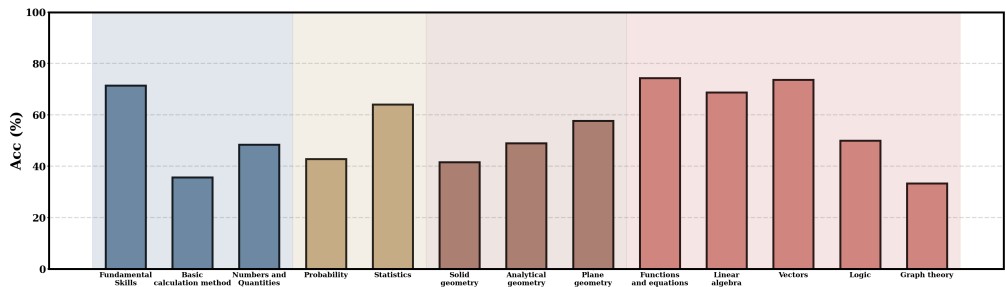

Figure 34: Detailed performance of Qwen2.5-VL-72B across 13 subdomains.

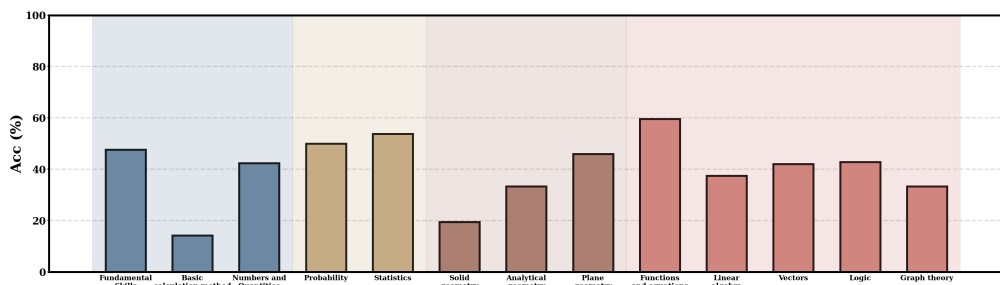

Figure 35: Detailed performance of LLaVA-OneVision-72B across 13 subdomains.

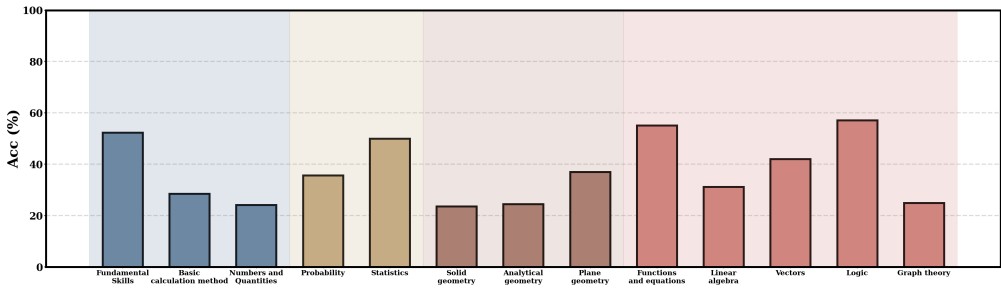

Figure 36: Detailed performance of InternVL2.5-8B across 13 subdomains.

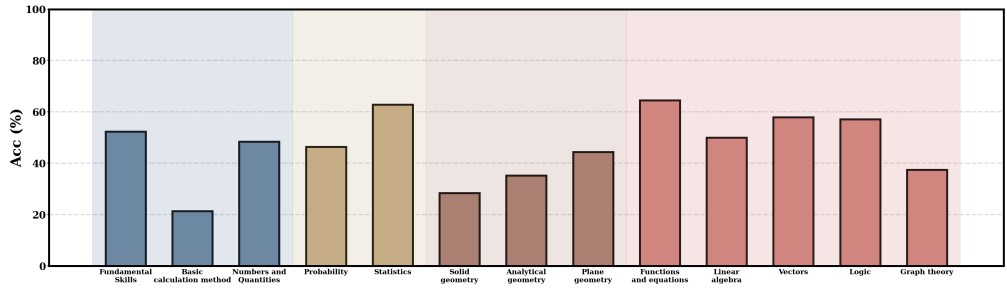

Figure 37: Detailed performance of Qwen2.5-VL-7B across 13 subdomains.

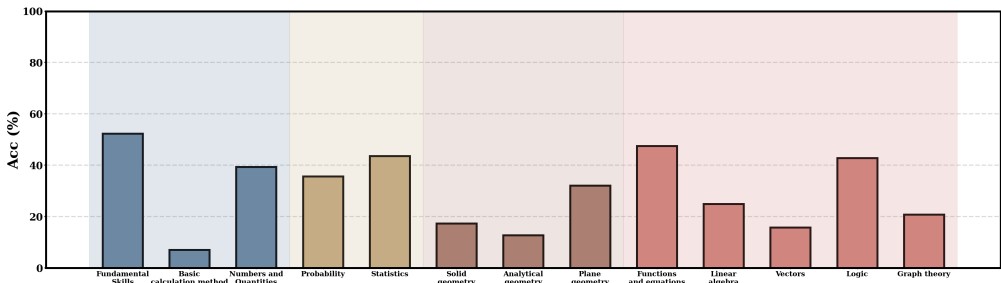

Figure 38: Detailed performance of LLaVA-OneVision-7B across 13 subdomains.

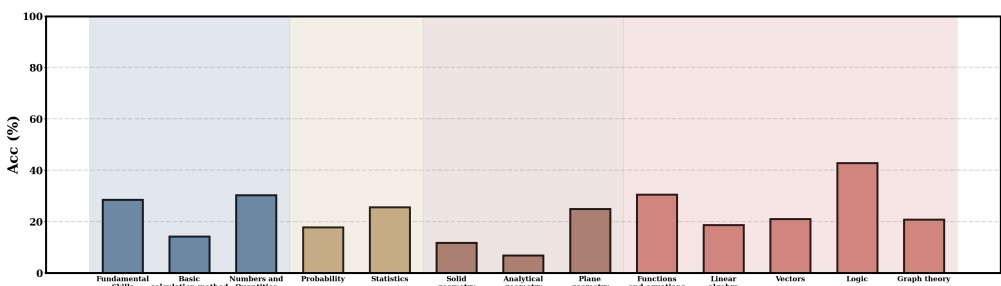

Figure 39: Detailed performance of GLM-4V-9B across 13 subdomains.

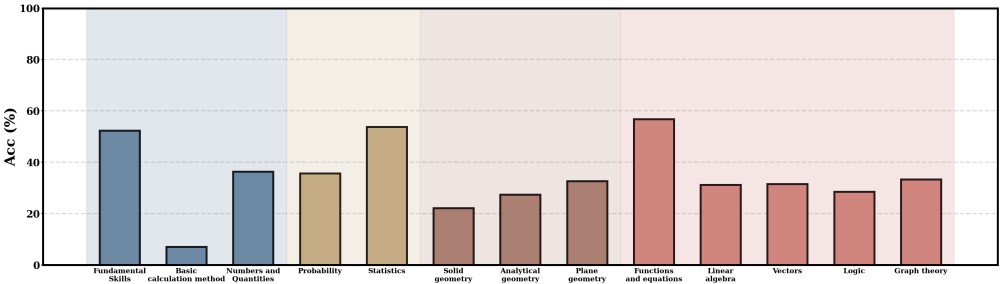

Figure 40: Detailed performance of Qwen2.5-VL-3B across 13 subdomains.

