# OpenReview forum: "We-Math 2.0: A Versatile MathBook System for Incentivizing Visual Mathematical Reasoning"
_ICLR.cc/2026/Conference — ICLR 2026 Poster_

### Official Review · Reviewer_K8f9 · 2025-10-30

**Soundness:** 3
**Presentation:** 3
**Contribution:** 3
**Rating:** 6
**Confidence:** 4

**Summary:**

The paper proposes a unified system to enhance the mathematical reasoning abilities of MLLMs. WE-MATH 2.0 combines a structured math knowledge base, a model-driven data design framework, and an RL-based learning curriculum. Results on various benchmarks suggest the effectiveness of the proposed framework for MLLM reasoning.

**Strengths:**

- WE-MATH2.0 Integrates data generation, training, and evaluation within a closed-loop system, ensuring consistent metrics between learning objectives and assessment benchmarks.

- The paper provides a large-scale, well-balanced benchmark covering both knowledge breadth and reasoning depth, enabling systematic evaluation of multimodal reasoning models.

**Weaknesses:**

- The performance improvements of MathBook-7B are mainly observed on its self-constructed benchmarks like We-Math, while gains on external benchmarks such as MathVista and MathVerse are relatively minor.
- Although MathBookEval is designed to test reasoning depth and knowledge coverage independently from the training data, MathBook-7B does not show significant advantages on this benchmark.
- The system focuses only on multimodal reasoning. It has not been evaluated on standard textual datasets such as GSM8K or MATH, leaving its language-only mathematical reasoning ability unverified. It would be great to also discuss the capability on language-only but still challenging math reasoning tasks.

**Questions:**

- How well does MathBook-7B generalize to language-only mathematical reasoning tasks, such as GSM8K or MATH, given that the system is trained and evaluated only on multimodal data?
- Have the authors conducted human validation or cross-checking to verify the correctness of automatically assigned knowledge labels and reasoning steps?
- Although MathBook-7B achieved strong results on We-Math and MathBookEval, how do the authors ensure that evaluation items were not seen or paraphrased during training, given the shared data construction pipeline?

---

> ### Author Response · Authors · 2025-11-23
> **Response to Reviewer K8f9 (1/2)**
>
> Dear Reviewer,
>
> We sincerely appreciate your recognition of We-Math 2.0 and your thoughtful comments and constructive feedback! Below we provide detailed responses to each of the raised concerns.
>
> ---
>
> > ### Q1 & W3: Generalization to language-only mathematical reasoning
>
>
> Thank you for the suggestion. Following your advice, we evaluated MathBook-7B on GSM8K[1] and MATH-500[2]. As shown in the table, MathBook-7B achieves consistent improvements over Qwen2.5-VL-7B on both benchmarks. These results suggest that the structured knowledge system in We-Math 2.0 contributes to generalizable reasoning ability across modalities.
>
> |Model|MATH-500|GSM8K|
> |---|---|---|
> |Qwen2.5-VL-7B|66.0|86.2|
> |MathBook-7B|67.2|87.6|
>
> ---
>
> > ### Q2: Human validation of knowledge labels and reasoning steps
>
>
> Thank you for highlighting this important aspect. Ensuring the correctness of knowledge annotations and reasoning steps is central to We-Math 2.0, and all components of the system have been cross-validated by human experts.
>
>
> - **For the MathBook Knowledge System:**
> As described in Lines 187–188:
> *“principles are consolidated and cross-checked with those written by human experts, with iterative refinement to ensure completeness and accuracy.”*
>
> - **For MathBookEval:**
> As stated in Lines 283–284, its construction involves:
> *“(2) expert step-by-step annotation with explicit knowledge-point mapping; and (3) independent cross-validation, retaining only consistently annotated items.”*
>
>
> - **For the training set:**
> All training problems are generated from the MathBook Knowledge System and refined by experts, ensuring consistency between the knowledge structure and the training data.
>
>
>
> ---
>
> > ### Q3: Ensuring that evaluation items were not seen or paraphrased during training
>
> Thank you for raising this important question. We confirm that no evaluation items in We-Math or MathBookEval appear in the training data, nor are any paraphrased variants included. Below we clarify why leakage cannot occur given the construction pipeline.
>
> - **Different data sources.**
> As described in Section 3.3 of We-Math[3], its problems are collected from publicly available mathematics websites, following the practices of MathVerse[4], MathVista[5], and MathVision[6]. In contrast, We-Math 2.0 adopts a *model-assisted, expert-led* workflow: given a knowledge point and its associated principles, an LLM first drafts a problem (problem, answer, and XML script), and human experts then refine and revise it. The two pipelines therefore originate from completely different sources.
>
> - **Different evaluation mechanisms.**
> We-Math is a process-based benchmark: each problem is decomposed into sub-questions, and a strict “all-correct” criterion is applied. We-Math 2.0 does not use this mechanism.
>
>
> - **Different knowledge scope.**
> We-Math focuses on elementary-level content. In contrast, We-Math 2.0 is built around a comprehensive knowledge system that spans advanced high-school topics, and competition-level mathematics. Its knowledge hierarchy and granularity differ substantially from We-Math: We-Math 2.0 is organized around *definitions, theorems,* and *applications*. As illustrated in Figure 1, the dataset is designed to cover a full spectrum of mathematical knowledge and to emphasize problems requiring deeper reasoning.
>
>
> In addition, your question also inspired us to explore the results from a broader perspective. We found that recent RL-based multimodal reasoning models (such as MM-Eureka[7], VLAA-Thinker[8], and WeThink[9]) consistently achieve substantial improvements on We-Math compared to earlier representative SFT-based models (for example, MathPuma[10] reports 19.2%). This trend suggests that the observed gains on We-Math are largely due to the rollout-based reasoning strategies introduced by RL (such as GRPO). The rollout mechanism enables models to explore multi-step solution paths during training, which naturally aligns with the process-oriented design of We-Math and leads to improved performance.

---

> ### Author Response · Authors · 2025-11-23
> **Response to Reviewer K8f9 (2/2)**
>
> ---
>
> > ### W1 & W2: Consistency of improvements across models and settings
>
> Thank you for pointing this out. In We-Math 2.0, we evaluate using both outcome-based and process-oriented benchmarks, and the system demonstrates consistent improvements over the baseline across these settings.
>
> To further address your concern, we additionally include:
> - results from multiple open-source reasoning models on MathBookEval,
>
> |Model|Acc.|Level1|Level2|Level3|FS.|PS.|Geo.|Alg.|
> |---|---|---|---|---|---|---|---|---|
> |MM-Eureka-7B[7]|50.0|50.6|51.5|38.9|47.1|52.8|44.0|62.9|
> |R1-VL-7B[11]|38.0|41.9|32.6|27.8|38.2|38.7|32.3|50.4|
> |OpenVLThinker-7B[12]|32.6|33.0|32.6|29.2|45.6|38.7|24.9|43.4|
> |VLAA-Thinker-7B[8]|35.7|38.3|31.9|29.2|46.6|42.5|30.2|41.6|
> |MathBook-7B|50.4|52.0|48.2|45.8|57.4|67.9|40.5|63.3|
>
> - results from MathBook-3B, a smaller backbone trained with the full pipeline,
>
> |Model|MathVista|MathVision|WeMath|MathVerse|MathBookEval|
> |---|---|---|---|---|---|
> |Qwen2.5-VL-3B|60.8|21.1|22.9|29.1|36.9|
> |MathBook-3B|63.2|24.3|28.1|32.6|40.2|
>
> - and the new text-only evaluations on GSM8K and MATH-500(from Q1).
>
> Across all of these newly added evaluations, We-Math 2.0 continues to show consistent improvements over the baselines, supporting the effectiveness of both the structured data design and the training methodology. All results will be included in the revised version.
>
> ---
> Thank you again for your thoughtful feedback and for engaging deeply with our work. We hope that the expanded analyses and additional experiments address your concerns. All corresponding results have been incorporated into the revised version of the paper ***(including Appendix C.3 Generalization to Text-Only Mathematical Reasoning Tasks, Appendix B.4.6 Additional Results on MathBookEval, and Appendix C.2 Additional Experimental Results for MathBook-3B)***.
>
> Thank you once again.
>
>
> ---
> > ### Reference
>
> [1] Training Verifiers to Solve Math Word Problems
>
>
> [2] Let's Verify Step by Step
>
>
> [3] We-Math: Does Your Large Multimodal Model Achieve Human-like Mathematical Reasoning?
>
>
> [4] MathVerse: Does Your Multi-modal LLM Truly See the Diagrams in Visual Math Problems?
>
>
> [5] MathVista: Evaluating Mathematical Reasoning of Foundation Models in Visual Contexts
>
>
> [6] Measuring Multimodal Mathematical Reasoning with MATH-Vision Dataset
>
>
> [7] MM-Eureka: Exploring the Frontiers of Multimodal Reasoning with Rule-based Reinforcement Learning
>
>
> [8] SFT or RL? An Early Investigation into Training R1-Like Reasoning Large Vision-Language Models
>
>
> [9] WeThink: Toward General-purpose Vision-Language Reasoning via Reinforcement Learning
>
>
> [10] Math-PUMA: Progressive Upward Multimodal Alignment to Enhance Mathematical Reasoning
>
>
> [11] R1-VL: Learning to Reason with Multimodal Large Language Models via Step-wise Group Relative Policy Optimization
>
>
> [12] OpenVLThinker: Complex Vision-Language Reasoning via Iterative SFT-RL Cycles

---

> > ### Comment · Reviewer_K8f9 · 2025-11-28
> > **Thank you for your reply**
> >
> > I thank the authors for their thorough response addressing my questions and concerns. I have no further questions at this time.

---

### Official Review · Reviewer_TdME · 2025-10-30

**Soundness:** 3
**Presentation:** 3
**Contribution:** 3
**Rating:** 8
**Confidence:** 3

**Summary:**

This paper introduces We-Math 2.0, a system involving a knowledge system for mathematical knowledge, a pair of dataset for training, an RL framework, and a comprehensive benchmark for multimodal mathematical reasoning.

**Strengths:**

1. This work represents substantial engineering effort in an important area and should be applauded for this.
2. MathBook Knowledge System (MKS) is comprehensive with 491 knowledge points + 1819 fundamental principles
3. MathBook-Standard/Pro is Built on MKS with annotated problems which are shown later in experiments to be strong together with the proposed MathBook-RL.
4. MathBook-RL is presented well with ablation studies.
5. This paper offers a few insights including the observation that MLLMs performance in geometry is subpar compared to that in algebra, and that performance in general correlates negatively with knowledge points.

**Weaknesses:**

Maybe some more experiments on a few more model scale are warranted, but considering the amount of work put into the entire system, I do not see this as much of defect.

**Questions:**

Do we have a mechanism to use the knowledge system at inference?

---

> ### Author Response · Authors · 2025-11-23
> **Response to Reviewer TdME**
>
> Dear Reviewer,
>
> We sincerely appreciate your recognition of We-Math 2.0, and thank you for raising these thoughtful questions! Below we provide a detailed and structured response.
>
>
> ---
> > ### W1: Additional experiments on model scaling
>
> Thank you for the suggestion. To further address your concern, we additionally trained a MathBook-3B model using the Qwen2.5-VL-3B as the backbone.
>
> Across MathVista[1], MathVision[2], MathVerse[3] and We-Math[4], MathBook-3B consistently outperforms the Qwen2.5-VL-3B:
>
> |Model|MathVista|MathVision|WeMath|MathVerse|MathBookEval|
> |---|---|---|---|---|---|
> |Qwen2.5-VL-3B|60.8|21.1|22.9|29.1|36.9|
> |MathBook-3B|63.2|24.3|28.1|32.6|40.2|
>
> These results further demonstrate that We-Math 2.0 provides high-quality and generalizable supervision, even at small model scales. **The corresponding results have been included in Appendix C.2 (Additional Experimental Results for MathBook-3B).**
>
>
>
> ---
> > ### Q1: Using the knowledge system at inference time
>
> Thank you for this insightful question. Motivated by your suggestion, we examined whether the structured knowledge system can be effectively leveraged during inference. In our experiments, ***knowledge-augmented inference*** refers to providing the structured knowledge annotations together with the input question.
>
> |Model|Acc.|Level1|Level2|Level3|Geo.|Alg.|
> |---|---|---|---|---|---|---|
> |Qwen2.5-VL-7B|46.7|50.1|43.0|33.3|38.8|60.2|
> |Qwen2.5-VL-7B +KA|48.9|50.1|49.2|37.5|39.8|63.4|
> |MathBook-7B|50.4|52.0|48.2|45.8|40.5|63.3|
> |MathBook-7B +KA|55.7|56.4|56.0|48.6|47.5|68.8|
>
> Based on these results, we summarize our findings below:
>
> - **Inference-time knowledge consistently improves both models.**
> Under ***knowledge-augmented inference***, Qwen2.5-VL-7B increases in accuracy from 46.7 to 48.9, and MathBook-7B increases from 50.4 to 55.7.
>
> - **MathBook-7B shows larger gains.**
> Experiments suggesting that systematic training with structured knowledge enables it to use explicit knowledge more effectively during inference.
>
> - **Lower-level knowledge (Level 1–2) provides the strongest improvements.**
> For MathBook-7B, Level 1 increases from 52.0 to 56.4, and Level 2 increases from 48.2 to 56.0, indicating that concise and highly relevant knowledge grounding is particularly beneficial.
>
> - **At Level 3, where the number of knowledge tags is substantially larger, the improvement is relatively modest.**
> This suggests that long knowledge sequences may dilute the effective signal or introduce token overhead, highlighting the need for more compact or efficient knowledge representations.
>
> Furthermore, your question also encouraged us to consider a more systematic inference mechanism. A direction is for the model to (1) identify the relevant knowledge concepts, (2) condense them into a compact internal representation, and (3) perform reasoning conditioned on this representation.
> We believe such a knowledge-aware inference process can further enhance both reliability and generalization.
>
>
>
> ---
>
> Thank you again for your constructive feedback and positive comments on We-Math 2.0. The above experiments have been incorporated into the revised version of the paper (***Appendix C.2 Additional Experimental Results for MathBook-3B***), and we hope these additions help address your question!
>
> Thank you once again.
>
> ---
> > ### Reference
>
> [1] MathVista: Evaluating Mathematical Reasoning of Foundation Models in Visual Contexts
>
>
> [2] Measuring Multimodal Mathematical Reasoning with MATH-Vision Dataset
>
>
> [3] MathVerse: Does Your Multi-modal LLM Truly See the Diagrams in Visual Math Problems?
>
>
> [4] We-Math: Does Your Large Multimodal Model Achieve Human-like Mathematical Reasoning?

---

### Official Review · Reviewer_ZPo5 · 2025-10-30

**Soundness:** 2
**Presentation:** 2
**Contribution:** 2
**Rating:** 6
**Confidence:** 4

**Summary:**

The paper proposes We-Math 2.0, a unified framework centered around a structured MathBook knowledge system with 491 knowledge points. It introduces associated datasets and the MathBookEval benchmark, and designs a reinforcement learning training strategy based on curriculum learning. The authors claim their trained model achieves a marginal performance advantage on several existing mathematical multimodal benchmarks

**Strengths:**

- The paper presents a unified framework (We-Math 2.0) that integrates a structured mathematical knowledge system, a model-centric data space, and an RL-based training paradigm.

- The trained model reportedly demonstrates a marginal advantage on some established mathematical multimodal benchmarks.

**Weaknesses:**

- The categorization and comparison in Table 1 appear inconsistent. For instance, comparing the granularity of the proposed 491-point system with datasets like MathV360k (which contains diverse content like charts and general QA) is not an apples-to-apples comparison and may be misleading.

**Questions:**

- How can the evaluation in Table 1 be made fair and consistent, especially regarding the granularity of category definitions?
- Could the evaluation on MathBookEval include more open-source reasoning models of comparable scale? This would help disentangle whether the observed performance gains primarily stem from the richness and structure of the training data or from the effectiveness of the MathBook-RL training pipeline itself.

---

> ### Author Response · Authors · 2025-11-23
> **Response to Reviewer ZPo5**
>
> Dear Reviewer,
>
> We sincerely appreciate your recognition of We-Math 2.0 and the thoughtful comments of our work! Your observations regarding Table 1 and the evaluation on MathBookEval are valuable, and we address them below.
>
> ---
>
> > ### Q1 & W1: Fairness and consistency in the presentation of Table 1
>
> Thank you for bringing this to our attention. Our initial design of Table 1 followed the style used in GeoSense[1]. However, as you noted, category definitions across existing datasets differ substantially in purpose and granularity, making strict alignment challenging. To present the comparison more clearly and avoid potential misinterpretation, we have revised Table 1 in the updated version.
>
> Specifically, instead of listing numerical counts, we now describe **whether each dataset includes explicit knowledge annotation and use checkmarks** rather than numbers to indicate its presence. Our intention is to clearly indicate that We-Math 2.0 provides structured knowledge annotations. We hope this revision presents the information in a clearer and more neutral manner.
>
>
> ---
>
> > ### Q2: Additional reasoning models in the MathBookEval evaluation
>
> Thank you for this helpful suggestion. We expanded the evaluation by adding several open-source reasoning models of comparable scale to MathBook-7B, covering different training paradigms. The new results show that **MathBook-7B remains competitive even with substantially less training data**, and its advantage is particularly clear on **Level 3** problems that require deeper reasoning. These findings support the contribution of both the structured design of We-Math 2.0 and the MathBook-RL training pipeline.
>
> |Model|Acc.|Level1|Level2|Level3|FS.|PS.|Geo.|Alg.|
> |---|---|---|---|---|---|---|---|---|
> |MM-Eureka-7B[2]|50.0|50.6|51.5|38.9|47.1|52.8|44.0|62.9|
> |R1-VL-7B[3]|38.0|41.9|32.6|27.8|38.2|38.7|32.3|50.4|
> |OpenVLThinker-7B[4]|32.6|33.0|32.6|29.2|45.6|38.7|24.9|43.4|
> |VLAA-Thinker-7B[5]|35.7|38.3|31.9|29.2|46.6|42.5|30.2|41.6|
> |MathBook-7B|50.4|52.0|48.2|45.8|57.4|67.9|40.5|63.3|
>
>
> Furthermore, we also provide results for **MathBook-3B**, a new smaller backbone trained with the same methodology. MathBook-3B consistently outperforms its baseline across multiple benchmarks. These additional evaluations further support the generality and effectiveness of our dataset and training framework.
>
> |Model|MathVista[6]|MathVision[7]|WeMath[8]|MathVerse[9]|MathBookEval|
> |---|---|---|---|---|---|
> |Qwen2.5-VL-3B|60.8|21.1|22.9|29.1|36.9|
> |MathBook-3B|63.2|24.3|28.1|32.6|40.2|
>
> ---
>
> We appreciate your constructive comments and your engagement with We-Math 2.0! All new results have been incorporated into the revised version of the paper ***(including updates to Table 1, Table 5, and Appendix B.4.6 Additional Results on MathBookEval)***. We hope that the updates and expanded experiments provide a clear response to your concerns.
>
> Thank you once again.
>
> ---
> > ### Reference
>
> [1] GeoSense: Evaluating Identification and Application of Geometric Principles in Multimodal Reasoning
>
>
> [2] MM-Eureka: Exploring the Frontiers of Multimodal Reasoning with Rule-based Reinforcement Learning
>
>
> [3] R1-VL: Learning to Reason with Multimodal Large Language Models via Step-wise Group Relative Policy Optimization
>
>
> [4] OpenVLThinker: Complex Vision-Language Reasoning via Iterative SFT-RL Cycles
>
>
> [5] SFT or RL? An Early Investigation into Training R1-Like Reasoning Large Vision-Language Models
>
>
> [6] MathVista: Evaluating Mathematical Reasoning of Foundation Models in Visual Contexts
>
>
> [7] Measuring Multimodal Mathematical Reasoning with MATH-Vision Dataset
>
>
> [8] We-Math: Does Your Large Multimodal Model Achieve Human-like Mathematical Reasoning?
>
>
> [9] MathVerse: Does Your Multi-modal LLM Truly See the Diagrams in Visual Math Problems?

---

### Official Review · Reviewer_FXnB · 2025-11-01

**Soundness:** 3
**Presentation:** 3
**Contribution:** 3
**Rating:** 6
**Confidence:** 3

**Summary:**

The authors proposed We-Math 2.0, a comprehensive framework for visual mathematical reasoning benchmarks. The major contribution of this paper is providing a detailed way to curate the dataset alongside with itself, paving the path for future progress. The authors also include a two-stage RL framework for MLLMs, and a extensive evaluation.

**Strengths:**

1. The paper is well structured and easy to follow. The teaser figure is particularly well structured.
2. The intuition of the dataset is detailed and inspiring, which could be even more helpful than the dataset iteself.
3. The evaluation results show the improvement brought by We-Math 2.0.

**Weaknesses:**

No significant weakness within the dataset scope.

**Questions:**

1. I see in the teaser figure the definition has been emphasized a lot. I wonder if the authors want to state that the solution of the question is related to the definition, or the question is about the definition itself?

---

> ### Author Response · Authors · 2025-11-23
> **Response to Reviewer FXnB**
>
> Dear Reviewer,
>
> We sincerely appreciate your recognition of We-Math 2.0 and appreciate your careful reading of the teaser figure! Below we clarify the purpose of emphasizing the “definition” element from two perspectives.
>
>
> ---
> > ### Q1: Clarification regarding the emphasis on “definition” in the teaser figure
>
> 1. **The “definition” in the teaser is part of the knowledge system, not the problem itself.**
> As described in Line 149, the knowledge hierarchy of We-Math 2.0 is built around *definitions, theorems,* and *applications*. Definition-level knowledge is particularly scarce in existing multimodal math datasets, so we intentionally highlight it in the teaser.
>    For instance, in the multiplication example, we visualize five groups of three (3 + 3 + 3 + 3 + 3) to make explicit the definition of multiplication as repeated addition, allowing the model to ground its understanding in the underlying concept rather than in surface-level formulas.
>
>
> 2. **This design also reflects our vision for knowledge-driven mathematical reasoning.**
>    Prior analyses suggest that existing models may sometimes generate answers based on memorized patterns rather than a principled understanding of the underlying knowledge-for example, achieving similar performance across elementary and high-school problems (MathVista[1]), or solving a full question while failing its decomposed sub-questions (We-Math[2]). These observations indicate that models may not always ground their reasoning in the appropriate underlying knowledge.
>    Our long-term goal is to encourage models to “learn from knowledge itself,” applying the correct concepts at the appropriate level of abstraction. The knowledge-centric design of We-Math 2.0, together with incorporating structured knowledge into the SFT cold-start, and constructing the MathBook-Standard and MathBook-Pro datasets for RL training, aims to support more principled reasoning, stronger generalization, and more reliable model behavior.
>
> ---
>
> We hope this clarifies the intention behind the teaser figure and provides additional insight into the knowledge-driven motivation of We-Math 2.0. Thank you again for your thoughtful question!
>
> Thank you once again.
>
> ---
> > ### Reference
>
> [1] MathVista: Evaluating Mathematical Reasoning of Foundation Models in Visual Contexts
>
>
> [2] We-Math: Does Your Large Multimodal Model Achieve Human-like Mathematical Reasoning?

---

### Meta-Review · Area_Chair_cnCe · 2026-01-07

**Summary:**

This paper presents We-Math 2.0, a comprehensive framework for multimodal mathematical reasoning that integrates a structured MathBook knowledge system, curated datasets, a curriculum-based RL training strategy, and a dedicated evaluation benchmark. Reviewers agree on the clarity, substantial engineering effort, and the lasting value of the dataset design and knowledge taxonomy. While gains on external benchmarks are sometimes modest, the overall contribution is strong, well validated, and provides a solid foundation for future research in multimodal mathematical reasoning.

**Reviewer Concerns:**

Reviewers are overall positive about this work.

**Reviewer Scores:**

Reviewers keep their positive scores after rebuttal.

---

### Decision · Program_Chairs · 2026-01-26

Accept (Poster)